# SURROGATE GAP MINIMIZATION IMPROVES SHARPNESS-AWARE TRAINING

**Juntang Zhuang**[1] [*]
j.zhuang@yale.edu

**Boqing Gong**[2]**, Liangzhe Yuan**[2]**, Yin Cui**[2]**, Hartwig Adam**[2]
{bgong, lzyuan, yincui, hadam}@google.com

**Nicha C. Dvornek**[1]**, Sekhar Tatikonda**[1]**, James S. Duncan**[1]
{nicha.dvornek, sekhar.tatikonda, james.duncan}@yale.edu

**Ting Liu**[2]
liuti@google.com

[1] Yale University, [2] Google Research

## ABSTRACT

The recently proposed Sharpness-Aware Minimization (SAM) improves generalization by minimizing a *perturbed loss* defined as the maximum loss within a neighborhood in the parameter space. However, we show that both sharp and flat minima can have a low perturbed loss, implying that SAM does not always prefer flat minima. Instead, we define a *surrogate gap*, a measure equivalent to the dominant eigenvalue of Hessian at a local minimum when the radius of neighborhood (to derive the perturbed loss) is small. The surrogate gap is easy to compute and feasible for direct minimization during training. Based on the above observations, we propose Surrogate **G**ap Guided **S**harpness-**A**ware **M**inimization (GSAM), a novel improvement over SAM with negligible computation overhead. Conceptually, GSAM consists of two steps: 1) a gradient descent like SAM to minimize the perturbed loss, and 2) an *ascent* step in the *orthogonal* direction (after gradient decomposition) to minimize the surrogate gap and yet not affect the perturbed loss. GSAM seeks a region with both small loss (by step 1) and low sharpness (by step 2), giving rise to a model with high generalization capabilities. Theoretically, we show the convergence of GSAM and provably better generalization than SAM. Empirically, GSAM consistently improves generalization (e.g., +3.2% over SAM and +5.4% over AdamW on ImageNet top-1 accuracy for ViT-B/32). Code is released at https://sites.google.com/view/gsam-iclr22/home.

## 1 INTRODUCTION

Modern neural networks are typically highly over-parameterized and easy to overfit to training data, yet the generalization performances on unseen data (test set) often suffer a gap from the training performance (Zhang et al., 2017a). Many studies try to understand the generalization of machine learning models, including the Bayesian perspective (McAllester, 1999; Neyshabur et al., 2017), the information perspective (Liang et al., 2019), the loss surface geometry perspective (Hochreiter & Schmidhuber, 1995; Jiang et al., 2019) and the kernel perspective (Jacot et al., 2018; Wei et al., 2019). Besides analyzing the properties of a model after training, some works study the influence of training and the optimization process, such as the implicit regularization of stochastic gradient descent (SGD) (Bottou, 2010; Zhou et al., 2020), the learning rate's regularization effect (Li et al., 2019), and the influence of the batch size (Keskar et al., 2016).

These studies have led to various modifications to the training process to improve generalization. Keskar & Socher (2017) proposed to use Adam in early training phases for fast convergence and then switch to SGD in late phases for better generalization. Izmailov et al. (2018) proposed to average weights to achieve a wider local minimum, which is expected to generalize better than sharp minima. A similar idea was later used in Lookahead (Zhang et al., 2019). Entropy-SGD (Chaudhari

---

[*]Work was done during an internship at Google

et al., 2019) derived the gradient of local entropy to avoid solutions in sharp valleys. Entropy-SGD has a nested Langevin iteration, inducing much higher computation costs than vanilla training.

The recently proposed Sharpness-Aware Minimization (SAM) (Foret et al., 2020) is a generic training scheme that improves generalization and has been shown especially effective for Vision Transformers (Dosovitskiy et al., 2020) when large-scale pre-training is unavailable (Chen et al., 2021). Suppose vanilla training minimizes loss $f(w)$ (e.g., the cross-entropy loss for classification), where $w$ is the parameter. SAM minimizes a *perturbed loss* defined as $f_p(w) \triangleq \max_{||\delta|| \leq \rho} f(w + \delta)$, which is the *maximum* loss within radius $\rho$ centered at the model parameter $w$. Intuitively, vanilla training seeks a single point with a low loss, while SAM searches for a neighborhood within which the maximum loss is low. However, we show that a low perturbed loss $f_p$ could appear in both flat and sharp minima, implying that only minimizing $f_p$ is not always sharpness-aware.

Although the perturbed loss $f_p(w)$ might disagree with sharpness, we find a *surrogate gap* defined as $h(w) \triangleq f_p(w) - f(w)$ agrees with sharpness — Lemma 3.3 shows that the surrogate gap $h$ is an equivalent measure of the dominant eigenvalue of Hessian at a local minimum. Inspired by this observation, we propose the Surrogate **G**ap Guided **S**harpness **A**ware **M**inimization (GSAM) which jointly minimizes the perturbed loss $f_p$ and the surrogate gap $h$: a low perturbed loss $f_p$ indicates a low training loss within the neighborhood, and a small surrogate gap $h$ avoids solutions in sharp valleys and hence narrows the generalization gap between training and test performances (Thm. 5.3). When both criteria are satisfied, we find a generalizable model with good performances.

GSAM consists of two steps for each update: 1) descend gradient $\nabla f_p(w)$ to minimize the perturbed loss $f_p$ (this step is exactly the same as SAM), and 2) decompose gradient $\nabla f(w)$ of the original loss $f(w)$ into components that are parallel and orthogonal to $\nabla f_p(w)$, i.e., $\nabla f(w) = \nabla_{\parallel} f(w) + \nabla_{\perp} f(w)$, and perform an ascent step in $\nabla_{\perp} f(w)$ to minimize the surrogate gap $h(w)$. Note that this ascent step does not change the perturbed loss $f_p$ because $\nabla f_{\perp}(w) \perp \nabla f_p(w)$ by construction.

We summarize our contribution as follows:

- We define *surrogate gap*, which measures the sharpness at local minima and is easy to compute.
- We propose the GSAM method to improve the generalization of neural networks. GSAM is widely applicable and incurs negligible computation overhead compared to SAM.
- We demonstrate the convergence of GSAM and its provably better generalization than SAM.
- We empirically validate GSAM over image classification tasks with various neural architectures, including ResNets (He et al., 2016), Vision Transformers (Dosovitskiy et al., 2020), and MLP-Mixers (Tolstikhin et al., 2021).

## 2 PRELIMINARIES

### 2.1 NOTATIONS

- $f(w)$: A loss function $f$ with parameter $w \in \mathbb{R}^k$, where $k$ is the parameter dimension.
- $\rho_t \in \mathbb{R}$: A scalar value controlling the amplitude of perturbation at step $t$.
- $\epsilon \in \mathbb{R}$: A small positive constant (to avoid division by 0, $\epsilon = 10^{-12}$ by default).
- $w_t^{adv} \triangleq w_t + \rho_t \frac{\nabla f(w_t)}{||\nabla f(w_t)|| + \epsilon}$: The solution to $\max_{||w' - w_t|| \leq \rho_t} f(w')$ when $\rho_t$ is small.
- $f_p(w_t) \triangleq \max_{||\delta|| \leq \rho_t} f(w_t + \delta) \approx f(w_t^{adv})$: The perturbed loss induced by $f(w_t)$. For each $w_t$, $f_p(w_t)$ returns the worst possible loss $f$ within a ball of radius $\rho_t$ centered at $w_t$. When $\rho_t$ is small, by Taylor expansion, the solution to the maximization problem is equivalent to a gradient ascent from $w_t$ to $w_t^{adv}$.
- $h(w) \triangleq f_p(w) - f(w)$: The surrogate gap defined as the difference between $f_p(w)$ and $f(w)$.
- $\eta_t \in \mathbb{R}$: Learning rate at step $t$.
- $\alpha \in \mathbb{R}$: A constant value that controls the scaled learning rate of the ascent step in GSAM.
- $g^{(t)}, g_p^{(t)} \in \mathbb{R}^k$: At the $t$-th step, the noisy observation of the gradients $\nabla f(w_t)$, $\nabla f_p(w_t)$ of the original loss and perturbed loss, respectively.

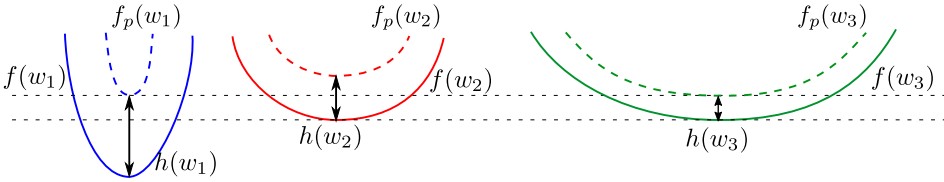

$$sharpness(w_1) > sharpness(w_2) > sharpness(w_3), \quad h(w_1) > h(w_2) > h(w_3),$$
$$f_p(w_1) = f_p(w_3) < f_p(w_2), \text{ hence } h \text{ agrees with sharpness while } f_p \text{ might not.}$$

Figure 1: Consider original loss $f$ (solid line), perturbed loss $f_p \triangleq \max_{||\delta|| \leq \rho} f(w+\delta)$ (dashed line), and surrogate gap $h(w) \triangleq f_p(w) - f(w)$. Intuitively, $f_p$ is approximately a max-pooled version of $f$ with a pooling kernel of width $2\rho$, and SAM minimizes $f_p$. From left to right are the local minima centered at $w_1, w_2, w_3$, and the valleys become flatter. Since $f_p(w_1) = f_p(w_3) < f_p(w_2)$, SAM prefers $w_1$ and $w_3$ to $w_2$. *However, a low $f_p$ could appear in both sharp ($w_1$) and flat ($w_3$) minima, so $f_p$ might disagree with sharpness.* On the contrary, a smaller surrogate gap $h$ indicates a flatter loss surface (Lemma 3.3). From $w_1$ to $w_3$, the loss surface is flatter, and $h$ is smaller.

- $\nabla f(w_t) = \nabla f_{||}(w_t) + \nabla f_{\perp}(w_t)$: Decompose $\nabla f(w_t)$ into parallel component $\nabla f_{||}(w_t)$ and vertical component $\nabla f_{\perp}(w_t)$ by projection $\nabla f(w_t)$ onto $\nabla f_p(w_t)$.

## 2.2 SHARPNESS-AWARE MINIMIZATION

Conventional optimization of neural networks typically minimizes the training loss $f(w)$ by gradient descent w.r.t. $\nabla f(w)$ and searches for a single point $w$ with a low loss. However, this vanilla training often falls into a sharp valley of the loss surface, resulting in inferior generalization performance (Chaudhari et al., 2019). Instead of searching for a single point solution, SAM seeks a region with low losses so that small perturbation to the model weights does not cause significant performance degradation. SAM formulates the problem as:

$$\min_w f_p(w) \text{ where } f_p(w) \triangleq \max_{||\delta|| \leq \rho} f(w + \delta) \tag{1}$$

where $\rho$ is a predefined constant controlling the radius of a neighborhood. This perturbed loss $f_p$ induced by $f(w)$ is the maximum loss within the neighborhood. When the perturbed loss is minimized, the neighborhood corresponds to low losses (below the perturbed loss). For a small $\rho$, using Taylor expansion around $w$, the inner maximization in Eq. 1 turns into a linear constrained optimization with solution

$$\arg\max_{||\delta|| \leq \rho} f(w + \delta) = \arg\max_{||\delta|| \leq \rho} f(w) + \delta^\top \nabla f(w) + O(\rho^2) = \rho \frac{\nabla f(w)}{||\nabla f(w)||} \tag{2}$$

As a result, the optimization problem of SAM reduces to

$$\min_w f_p(w) \approx \min_w f(w^{adv}) \text{ where } w^{adv} \triangleq w + \rho \frac{\nabla f(w)}{||\nabla f(w)|| + \epsilon} \tag{3}$$

where $\epsilon$ is a scalar (default: 1e-12) to avoid division by 0, and $w^{adv}$ is the "perturbed weight" with the highest loss within the neighborhood. Equivalently, SAM seeks a solution on the surface of the perturbed loss $f_p(w)$ rather than the original loss $f(w)$ (Foret et al., 2020).

## 3 THE SURROGATE GAP MEASURES THE SHARPNESS AT A LOCAL MINIMUM

### 3.1 THE PERTURBED LOSS IS NOT ALWAYS SHARPNESS-AWARE

Despite that SAM searches for a region of low losses, we show that a solution by SAM is not guaranteed to be flat. Throughout this paper we measure the sharpness at a local minimum of loss $f(w)$ by the dominant eigenvalue $\sigma_{max}$ (eigenvalue with the largest absolute value) of Hessian. For simplicity, we do not consider the influence of reparameterization on the geometry of loss surfaces, which is thoroughly discussed in (Laurent & Massart, 2000; Kwon et al., 2021).

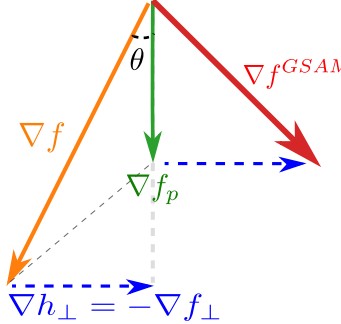

Figure 2: $\nabla f$ is decomposed into parallel and vertical ($\nabla f_\perp$) components by projection onto $\nabla f_p$. $\nabla f^{GSAM} = \nabla f_p - \alpha \nabla f_\perp$

**Algorithm 1** GSAM Algorithm

**For** $t = 1$ to $T$

0) $\rho_t$ schedule: $\rho_t = \rho_{min} + \frac{(\rho_{max} - \rho_{min})(lr - lr_{min})}{lr_{max} - lr_{min}}$

1a) $\Delta w_t = \rho_t \frac{\nabla f^{(t)}}{||\nabla f^{(t)}|| + \epsilon}$

1b) $w_t^{adv} = w_t + \Delta w_t$

2) Get $\nabla f_p^{(t)}$ by back-propagation at $w_t^{adv}$.

3) $\nabla f^{(t)} = \nabla f_\parallel^{(t)} + \nabla f_\perp^{(t)}$ Decompose $\nabla f^{(t)}$ into components that are parallel and orthogonal to $\nabla f_p^{(t)}$.

4) Update weights:

| | |
|---|---|
| Vanilla | $w_{t+1} = w_t - \eta_t \nabla f^{(t)}$ |
| SAM | $w_{t+1} = w_t - \eta_t \nabla f_p^{(t)}$ |
| GSAM | $w_{t+1} = w_t - \eta_t (\nabla f_p^{(t)} - \alpha \nabla f_\perp^{(t)})$ |

**Lemma 3.1.** *For some fixed $\rho$, consider two local minima $w_1$ and $w_2$, $f_p(w_1) \leq f_p(w_2) \;\not\Longrightarrow\; \sigma_{max}(w_1) \leq \sigma_{max}(w_2)$, where $\sigma_{max}$ is the dominant eigenvalue of the Hessian.*

We leave the proof to Appendix. Fig. 1 illustrates Lemma 3.1 with an example. Consider three local minima denoted as $w_1$ to $w_3$, and suppose the corresponding loss surfaces are flatter from $w_1$ to $w_3$. For some fixed $\rho$, we plot the perturbed loss $f_p$ and surrogate gap $h \triangleq f_p - f$ around each solution. Comparing $w_2$ with $w_3$: Suppose their vanilla losses are equal, $f(w_2) = f(w_3)$, then $f_p(w_2) > f_p(w_3)$ because the loss surface is flatter around $w_3$, implying that SAM will prefer $w_3$ to $w_2$. Comparing $w_1$ and $w_2$: $f_p(w_1) < f_p(w_2)$, and SAM will favor $w_1$ over $w_2$ because it only cares about the perturbed loss $f_p$, even though the loss surface is sharper around $w_1$ than $w_2$.

## 3.2 The Surrogate Gap Agrees with Sharpness

We introduce the surrogate gap that agrees with sharpness, defined as:

$$h(w) \triangleq \max_{||\delta|| \leq \rho} f(w + \delta) - f(w) \approx f(w^{adv}) - f(w) \tag{4}$$

Intuitively, the surrogate gap represents the difference between the maximum loss within the neighborhood and the loss at the center point. The surrogate gap has the following properties.

**Lemma 3.2.** *Suppose the perturbation amplitude $\rho$ is sufficiently small, then the approximation to the surrogate gap in Eq. 4 is always non-negative, $h(w) \approx f(w^{adv}) - f(w) \geq 0, \forall w$.*

**Lemma 3.3.** *For a local minimum $w^*$, consider the dominate eigenvalue $\sigma_{max}$ of the Hessian of loss $f$ as a measure of sharpness. Considering the neighborhood centered at $w^*$ with a small radius $\rho$, the surrogate gap $h(w^*)$ is an equivalent measure of the sharpness: $\sigma_{max} \approx 2h(w^*)/\rho^2$.*

The proof is in Appendix. Lemma 3.2 tells that the surrogate gap is non-negative, and Lemma 3.3 shows that the loss surface is flatter as $h$ gets closer to 0. The two lemmas together indicate that we can find a region with a flat loss surface by minimizing the surrogate gap $h(w)$.

# 4 Surrogate Gap Guided Sharpness-Aware Minimization

## 4.1 General idea: Simultaneously minimize the perturbed loss and surrogate gap

Inspired by the analysis in Section 3, we propose Surrogate **G**ap Guided **S**harpness-**A**ware **M**inimzation (GSAM) to simultaneously minimize two objectives, the perturbed loss $f_p$ and the surrogate gap $h$:

$$\min_w \big( f_p(w), h(w) \big) \tag{5}$$

Intuitively, by minimizng $f_p$ we search for a region with a low perturbed loss similar to SAM, and by minimizing $h$ we search for a local minimum with a flat surface. A low perturbed loss implies

low training losses within the neighborhood, and a flat loss surface reduces the generalization gap between training and test performances (Chaudhari et al., 2019). When both are minimized, the solution gives rise to high accuracy and good generalization.

**Potential caveat in optimization** It is tempting and yet sub-optimal to combine the objectives in Eq. 5 to arrive at $\min_w f_p(w) + \lambda h(w)$, where $\lambda$ is some positive scalar. One caveat when solving this weighted combination is the potential conflict between the gradients of the two terms, i.e., $\nabla f_p(w)$ and $\nabla h(w)$. We illustrate this conflict by Fig. 2, where $\nabla h(w) = \nabla f_p(w) - \nabla f(w)$ (the grey dashed arrow) has a negative inner product with $\nabla f_p(w)$ and $\nabla f(w)$. Hence, the gradient descent for the surrogate gap could potentially increase the loss $f_p$, harming the model's performance. We empirically validate this argument in Sec. 6.4.

## 4.2 GRADIENT DECOMPOSITION AND ASCENT FOR THE MULTI-OBJECTIVE OPTIMIZATION

Our primary goal is to minimize $f_p$ because otherwise a flat solution of high loss is meaningless, and the minimization of $h$ should not increase $f_p$. We propose to decompose $\nabla f(w_t)$ and $\nabla h$ into components that are parallel and orthogonal to $\nabla f_p(w_t)$, respectively (see Fig. 2):

$$
\begin{aligned}
\nabla f(w_t) &= \nabla f_{\|}(w_t) + \nabla f_{\perp}(w_t) \\
\nabla h(w_t) &= \nabla h_{\|}(w_t) + \nabla h_{\perp}(w_t) \\
\nabla h_{\perp}(w_t) &= -\nabla f_{\perp}(w_t)
\end{aligned}
\tag{6}
$$

The key is that updating in the direction of $\nabla h_{\perp}(w_t)$ does *not* change the value of the perturbed loss $f_p(w_t)$ because $\nabla h_{\perp} \perp \nabla f_p$ by construction. Therefore, we propose to perform a **descent step in the $\nabla h_{\perp}(w_t)$ direction**, which is equivalent to an **ascent step in the $\nabla f_{\perp}(w_t)$ direction** (because $\nabla h_{\perp} = -\nabla f_{\perp}$ by the definition of $h$), and achieve two goals simultaneously — it keeps the value of $f_p(w_t)$ intact and meanwhile decreases the surrogate gap $h(w_t) = f_p(w_t) - f(w_t)$ (by increasing $f(w_t)$ and not affect $f_p(w_t)$).

**The full GSAM Algorithm** is shown in Algo. 1 and Fig. 2, where $g^{(t)}, g_p^{(t)}$ are noisy observations of $\nabla f(w_t)$ and $\nabla f_p(w_t)$, respectively, and $g_{\|}^{(t)}, g_{\perp}^{(t)}$ are noisy observations of $\nabla f_{\|}(w_t)$ and $\nabla f_{\perp}(w_t)$, respectively, by projecting $g^{(t)}$ onto $g_p^{(t)}$. We introduce a constant $\alpha$ to scale the stepsize of the ascent step. Steps 1) to 2) are the same as SAM: At current point $w_t$, step 1) takes a gradient ascent to $w_t^{adv}$ followed by step 2) evaluating the gradient $g_p^{(t)}$ at $w_t^{adv}$. Step 3) projects $g^{(t)}$ onto $g_p^{(t)}$, which requires negligible computation compared to the forward and backward passes. In step 4), $-\eta_t g_p^{(t)}$ is the same as in SAM and minimizes the perturbed loss $f_p(w_t)$ with gradient descent, and $\alpha \eta_t g_{\perp}^{(t)}$ performs an *ascent* step in the orthogonal direction of $g_p^{(t)}$ to minimize the surrogate gap $h(w_t)$ ( equivalently increase $f(w_t)$ and keep $f_p(w_t)$ intact). In coding, GSAM feeds the "surrogate gradient" $\nabla f_t^{GSAM} \triangleq g_p^{(t)} - \alpha g_{\perp}^{(t)}$ to first-order gradient optimizers such as SGD and Adam.

**The ascent step along $g_{\perp}^{(t)}$ does not harm convergence** SAM demonstrates that minimizing $f_p$ makes the network generalize better than minimizing $f$. Even though our ascent step along $g_{\perp}^{(t)}$ increases $f(w)$, it does not affect $f_p(w)$, so GSAM still decreases the perturbed loss $f_p$ in a way similar to SAM. In Thm. 5.1, we formally prove the convergence of GSAM. In Sec. 6 and Appendix C, we empirically validate that the loss decreases and accuracy increases with training.

**Illustration with a toy example** We demonstrate different algorithms by a numerical toy example shown in Fig. 3. The trajectory of GSAM is closer to the ridge and tends to find a flat minimum. Intuitively, since the loss surface is smoother along the ridge than in sharp local minima, the surrogate gap $h(w)$ is small near the ridge, and the ascent step in GSAM minimizes $h$ to pushes the trajectory closer to the ridge. More concretely, $\nabla f(w_t)$ points to a sharp local solution and deviates from the ridge; in contrast, $w_t^{adv}$ is closer to the ridge and $\nabla f(w_t^{adv})$ is closer to the ridge descent direction than $\nabla f(w_t)$. Note that $\nabla f_t^{GSAM}$ and $\nabla f(w_t)$ always lie at different sides of $\nabla f_p(w_t)$ by construction (see Fig. 2), hence $\nabla f_t^{GSAM}$ pushes the trajectory closer to the ridge than $\nabla f_p(w_t)$ does. The trajectory of GSAM is like descent along the ridge and tends to find flat minima.

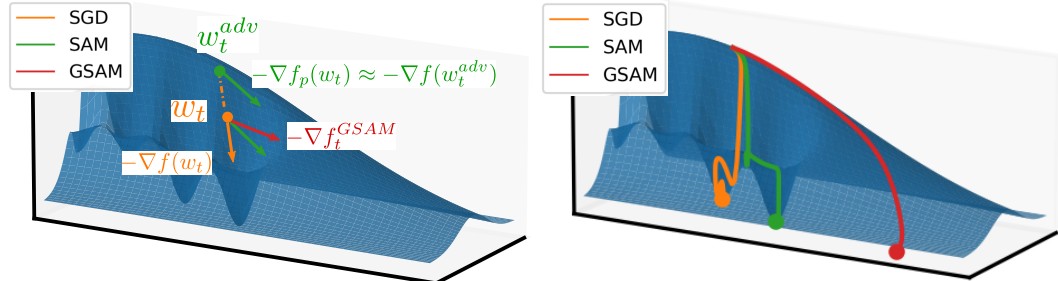

Figure 3: Consider the loss surface with a few sharp local minima. **Left**: Overview of the procedures of SGD, SAM and GSAM. SGD takes a descent step at $w_t$ using $\nabla f(w_t)$ (orange), which points to a sharp local minima. SAM first performs gradient ascent in the direction of $\nabla f(w_t)$ to reach $w_t^{adv}$ with a higher loss, followed by descent with gradient $\nabla f(w_t^{adv})$ (green) at the perturbed weight. Based on $\nabla f(w_t)$ and $\nabla f(w_t^{adv})$, GSAM updates in a new direction (red) that points to a flatter region. **Right**: Trajectories by different methods. SGD and SAM fall into different sharp local minima, while GSAM reaches a flat region. A video is in the supplement for better visualization.

## 5 THEORETICAL PROPERTIES OF GSAM

### 5.1 CONVERGENCE DURING TRAINING

**Theorem 5.1.** *Consider a non-convex function $f(w)$ with Lipschitz-smooth constant $L$ and lower bound $f_{min}$. Suppose we can access a noisy, bounded observation $g^{(t)}$ ($||g^{(t)}||_2 \leq G, \forall t$) of the true gradient $\nabla f(w_t)$ at the $t$-th step. For some constant $\alpha$, with learning rate $\eta_t = \eta_0/\sqrt{t}$, and perturbation amplitude $\rho_t$ proportional to the learning rate, e.g., $\rho_t = \rho_0/\sqrt{t}$, we have*

$$\frac{1}{T}\sum_{t=1}^{T}\mathbb{E}\Big|\Big|\nabla f_p(w_t)\Big|\Big|_2^2 \leq \frac{C_1 + C_2\log T}{\sqrt{T}}, \quad \frac{1}{T}\sum_{t=1}^{T}\mathbb{E}\Big|\Big|\nabla f(w_t)\Big|\Big|_2^2 \leq \frac{C_3 + C_4\log T}{\sqrt{T}}$$

*where $C_1, C_2, C_3, C_4$ are some constants.*

Thm. 5.1 implies both $f_p$ and $f$ converge in GSAM at rate $O(\log T/\sqrt{T})$ for non-convex stochastic optimization, matching the convergence rate of first-order gradient optimizers like Adam.

### 5.2 GENERALIZATION OF GSAM

In this section, we show the surrogate gap in GSAM is provably lower than SAM's, so GSAM is expected to find a smoother minimum with better generalization.

**Theorem 5.2** (PAC-Bayesian Theorem (McAllester, 2003)). *Suppose the training set has $m$ elements drawn i.i.d. from the true distribution, and denote the loss on the training set as $\widehat{f}(w) = \frac{1}{m}\sum_{i=1}^{m} f(w, x_i)$, where we use $x_i$ to denote the (input, target) pair of the $i$-th element. Let $w$ be learned from the training set. Suppose $w$ is drawn from posterior distribution $\mathcal{Q}$. Denote the prior distribution (independent of training) as $\mathcal{P}$, then*

$$\mathbb{E}_{w\sim\mathcal{Q}}\mathbb{E}_x f(w, x) \leq \mathbb{E}_{w\sim\mathcal{Q}}\widehat{f}(w) + 4\sqrt{\Big(KL(\mathcal{Q}||\mathcal{P}) + \log\frac{2m}{a}\Big)/m} \text{ with probability at least } 1 - a$$

**Corollary 5.2.1.** *Suppose perturbation $\delta$ is drawn from distribution $\delta \sim \mathcal{N}(0, b^2 I^k), \delta \in \mathbb{R}^k$, $k$ is the dimension of $w$, then with probability at least $\left(1 - a\right)\left[1 - e^{-\left(\frac{\rho}{\sqrt{2}b} - \sqrt{k}\right)^2}\right]$*

$$\mathbb{E}_{w\sim\mathcal{Q}}\mathbb{E}_x f(w, x) \leq \widehat{h} + C + 4\sqrt{\Big(KL(\mathcal{Q}||\mathcal{P}) + \log\frac{2m}{a}\Big)/m} \tag{7}$$

$$\widehat{h} \triangleq \max_{||\delta||_2 \leq \rho}\widehat{f}(w + \delta) - \widehat{f}(w) = \frac{1}{m}\sum_{i=1}^{m}\Big[\max_{||\delta||_2 \leq \rho} f(w + \delta, x_i) - f(w, x_i)\Big] \tag{8}$$

*where $C = \widehat{f}(w)$ is the empirical training loss, and $\widehat{h}$ is the surrogate gap evaluated on the training set.*

Corollary 5.2.1 implies that minimizing $\widehat{h}$ (right hand side of Eq. 7) is expected to achieve a tighter upper bound of the generalization performance (left hand side of Eq. 7). The third term on the right of Eq. 7 is typically hard to analyze and often simplified to $L2$ regularization (Foret et al., 2020). Note that $f_p = C + \widehat{h}$ only holds when $\rho_{train}$ (the perturbation amplitude specified by users during training) equals $\rho_{true}$ (the ground truth value determined by underlying data distribution); when $\rho_{train} \neq \rho_{true}$, $min(f_p, \widehat{h})$ is more effective than $min(f_p)$ in terms of minimizing generalization loss. A detailed discussion is in Appendix A.7.

**Theorem 5.3** (Unlike SAM, GSAM decreases the surrogate gap). *Under the assumption in Thm. 5.1, Thm. 5.2 and Corollary 5.2.1, we assume the Hessian has a lower-bound $|\sigma|_{min}$ on the absolute value of eigenvalue, and the variance of noisy observation $g^{(t)}$ is lower-bounded by $c^2$. The surrogate gap $h$ can be minimized by the ascent step along the orthogonal direction $g_{\perp}^{(t)}$. During training we minimize the sample estimate of $h$. We use $\Delta \widehat{h}_t$ to denote the amount that the ascent step in GSAM* **decreases** $\widehat{h}$ *for the $t$-th step. Compared to SAM, the proposed method generates a total decrease in surrogate gap $\sum_{t=1}^{T} \Delta \widehat{h}_t$, which is bounded by*

$$\frac{\alpha c^2 \rho_0^2 \eta_0 |\sigma|_{min}^2}{G^2} \leq \lim_{T \to \infty} \sum_{t=1}^{T} \Delta \widehat{h}_t \leq 2.7 \alpha L^2 \eta_0 \rho_0^2 \tag{9}$$

We provide proof in the appendix. The lower-bound of $\sum_{t=1}^{T} \Delta \widehat{h}_t$ indicates that GSAM achieves a provably non-trivial decrease in the surrogate gap. Combined with Corollary 5.2.1, GSAM provably improves the generalization performance over SAM.

## 6 EXPERIMENTS

### 6.1 GSAM IMPROVES TEST PERFORMANCE ON VARIOUS MODEL ARCHITECTURES

We conduct experiments with ResNets (He et al., 2016), Vision Transformers (ViTs) (Dosovitskiy et al., 2020) and MLP-Mixers (Tolstikhin et al., 2021). Following the settings by Chen et al. (2021), we train on the ImageNet-1k (Deng et al., 2009) training set using the Inception-style (Szegedy et al., 2015) pre-processing without extra training data or strong augmentation. For all models, we search for the best learning rate and weight decay for vanilla training, and then use the same values for the experiments with SAM and GSAM. For ResNets, we search for $\rho$ from 0.01 to 0.05 with a stepsize 0.01. For ViTs and Mixers, we search for $\rho$ from 0.05 to 0.6 with a stepsize 0.05. In GSAM, we search for $\alpha$ in $\{0.01, 0.02, 0.03\}$ for ResNets and $\alpha$ in $\{0.1, 0.2, 0.3\}$ for ViTs and Mixers. Considering that each step in SAM and GSAM requires twice the computation of vanilla training, we experiment with the vanilla training for twice the epochs of SAM and GSAM, but we observe no significant improvements from the longer training (Table 5 in appendix). We summarize the best hyper-parameters for each model in Appendix B.

We report the performances on ImageNet (Deng et al., 2009), ImageNet-v2 (Recht et al., 2019) and ImageNet-Real (Beyer et al., 2020) in Table 1. GSAM consistently improves over SAM and vanilla training (with SGD or AdamW): on ViT-B/32, GSAM achieves +5.4% improvement over AdamW and +3.2% over SAM in top-1 accuracy; on Mixer-B/32, GSAM achieves +11.1% over AdamW and +1.2% over SAM. We ignore the standard deviation since it is typically negligible ($< 0.1\%$) compared to the improvements. We also test the generalization performance on out-of-distribution data (ImageNet-R and ImageNet-C), and the observation is consistent with that on ImageNet, e.g., +5.1% on ImageNet-R and +5.9% on ImageNet-C for Mixer-B/32.

### 6.2 GSAM FINDS A MINIMUM WHOSE HESSIAN HAS SMALL DOMINANT EIGENVALUES

Lemma 3.3 indicates that the surrogate gap $h$ is an equivalent measure of the dominant eigenvalue of the Hessian, and minimizing $h$ equivalently searches for a flat minimum. We empirically validate this in Fig. 4. As shown in the left subfigure, for some fixed $\rho$, increasing $\alpha$ decreases the dominant value and improves generalization (test accuracy). In the middle subfigure, we plot the dominant

Table 1: Top-1 Accuracy (%) on ImageNet datasets for ResNets, ViTs and MLP-Mixers trained with Vanilla SGD or AdamW, SAM, and GSAM optimizers.

| Model | Training | ImageNet-v1 | ImageNet-Real | ImageNet-V2 | ImageNet-R | ImageNet-C |
|---|---|---|---|---|---|---|
| | | ResNet | | | | |
| ResNet50 | Vanilla (SGD) | 76.0 | 82.4 | 63.6 | 22.2 | 44.6 |
| | SAM | 76.9 | 83.3 | 64.4 | **23.8** | 46.5 |
| | **GSAM** | **77.2** | **83.9** | **64.6** | 23.6 | **47.6** |
| ResNet101 | Vanilla (SGD) | 77.8 | 83.9 | 65.3 | 24.4 | 48.5 |
| | SAM | 78.6 | 84.8 | 66.7 | 25.9 | 51.3 |
| | **GSAM** | **78.9** | **85.2** | **67.3** | **26.3** | **51.8** |
| ResNet152 | Vanilla (SGD) | 78.5 | 84.2 | 66.3 | 25.3 | 50.0 |
| | SAM | 79.3 | 84.9 | 67.3 | 25.7 | 52.2 |
| | **GSAM** | **80.0** | **85.9** | **68.6** | **27.3** | **54.1** |
| | | Vision Transformer | | | | |
| ViT-S/32 | Vanilla (AdamW) | 68.4 | 75.2 | 54.3 | 19.0 | 43.3 |
| | SAM | 70.5 | 77.5 | 56.9 | 21.4 | 46.2 |
| | **GSAM** | **73.8** | **80.4** | **60.4** | **22.5** | **48.2** |
| ViT-S/16 | Vanilla (AdamW) | 74.4 | 80.4 | 61.7 | 20.0 | 46.5 |
| | SAM | 78.1 | 84.1 | 65.6 | 24.7 | 53.0 |
| | **GSAM** | **79.5** | **85.3** | **67.3** | **25.3** | **53.3** |
| ViT-B/32 | Vanilla (AdamW) | 71.4 | 77.5 | 57.5 | 23.4 | 44.0 |
| | SAM | 73.6 | 80.3 | 60.0 | 24.0 | 50.7 |
| | **GSAM** | **76.8** | **82.7** | **63.0** | **25.1** | **51.7** |
| ViT-B/16 | Vanilla (AdamW) | 74.6 | 79.8 | 61.3 | 20.1 | 46.6 |
| | SAM | 79.9 | 85.2 | 67.5 | 26.4 | **56.5** |
| | **GSAM** | **81.0** | **86.5** | **69.2** | **27.1** | 55.7 |
| | | MLP-Mixer | | | | |
| Mixer-S/32 | Vanilla (AdamW) | 63.9 | 70.3 | 49.5 | 16.9 | 35.2 |
| | SAM | 66.7 | 73.8 | 52.4 | 18.6 | 39.3 |
| | **GSAM** | **68.6** | **75.8** | **55.0** | **22.6** | **44.6** |
| Mixer-S/16 | Vanilla (AdamW) | 68.8 | 75.1 | 54.8 | 15.9 | 35.6 |
| | SAM | 72.9 | 79.8 | 58.9 | 20.1 | 42.0 |
| | **GSAM** | **75.0** | **81.7** | **61.9** | **23.7** | **48.5** |
| Mixer-S/8 | Vanilla (AdamW) | 70.2 | 76.2 | 56.1 | 15.4 | 34.6 |
| | SAM | 75.9 | 82.5 | 62.3 | 20.5 | 42.4 |
| | **GSAM** | **76.8** | **83.4** | **64.0** | **24.6** | **47.8** |
| Mixer-B/32 | Vanilla (AdamW) | 62.5 | 68.1 | 47.6 | 14.6 | 33.8 |
| | SAM | 72.4 | 79.0 | 58.0 | 22.8 | 46.2 |
| | **GSAM** | **73.6** | **80.2** | **59.9** | **27.9** | **52.1** |
| Mixer-B/16 | Vanilla (AdamW) | 66.4 | 72.1 | 50.8 | 14.5 | 33.8 |
| | SAM | 77.4 | 83.5 | 63.9 | 24.7 | 48.8 |
| | **GSAM** | **77.8** | **84.0** | **64.9** | **28.3** | **54.4** |

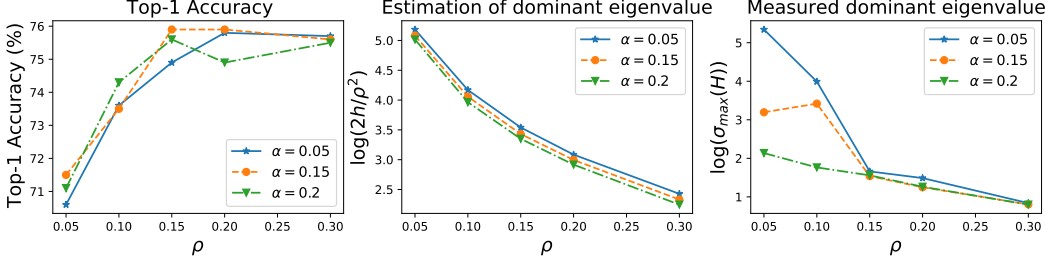

Figure 4: Influence of $\rho$ (set as constant for ease of comparison, other experiments use decayed $\rho_t$ schedule) and $\alpha$ on the training of ViT-B/32. **Left**: Top-1 accuracy on ImageNet. **Middle**: Estimation of the dominant eigenvalues from the surrogate gap, $\sigma_{max} \approx 2h/\rho^2$. **Right**: Dominant eigenvalues of the Hessian calculated via the power iteration. Middle and right figures match in the trend of curves, validating that the surrogate gap can be viewed as a proxy of the dominant eigenvalue of Hessian.

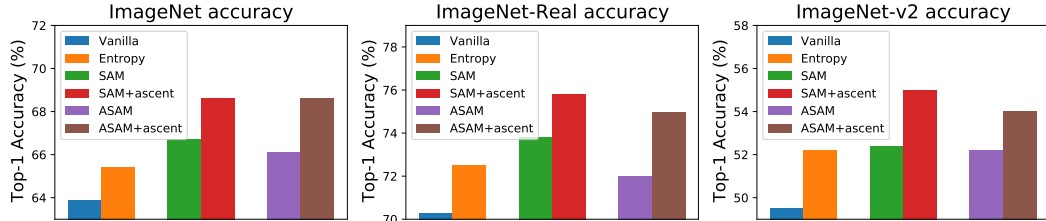

Figure 5: Top-1 accuracy of Mixer-S/32 trained with different methods. "+ascent" represents applying the ascent step in Algo. 1 to an optimizer. Note that our GSAM is described as SAM+ascent(**=GSAM**) for consistency.

Table 2: Results (%) of GSAM and $\min(f_p + \lambda h)$ on ViT-B/32

| Dataset | $\min(f_p + \lambda h)$ | GSAM |
|---|---|---|
| ImageNet | 75.4 | **76.8** |
| ImageNet-Real | 81.1 | **82.7** |
| ImageNet-v2 | 60.9 | **63.0** |
| ImageNet-R | 23.9 | **25.1** |

Table 3: Transfer learning results (top-1 accuracy, %)

| | ViT-B/16 | | | ViT-S/16 | | |
|---|---|---|---|---|---|---|
| | Vanilla | SAM | GSAM | Vanilla | SAM | GSAM |
| Cifar10 | 98.1 | 98.6 | **98.8** | 97.6 | 98.2 | **98.4** |
| Cifar100 | 87.6 | 89.1 | **89.7** | 85.7 | 87.6 | **88.1** |
| Flowers | 88.5 | **91.8** | 91.2 | 86.4 | **91.5** | 90.3 |
| Pets | 91.9 | 93.1 | **94.4** | 90.4 | 92.9 | **93.5** |
| mean | 91.5 | 93.2 | **93.5** | 90.0 | **92.6** | **92.6** |

eigenvalues estimated by the surrogate gap, $\sigma_{max} \approx 2h/\rho^2$ (Lemma 3.3). In the right subfigure, we directly calculate the dominant eigenvalues using the power-iteration (Mises & Pollaczek-Geiringer, 1929). The estimated dominant eigenvalues (middle) match the real eigenvalues $\sigma_{max}$ (right) in terms of the trend that $\sigma_{max}$ decreases with $\alpha$ and $\rho$. Note that the surrogate gap $h$ is derived over the whole training set, while the measured eigenvalues are over a subset to save computation. These results show that the ascent step in GSAM minimizes the dominant eigenvalue by minimizing the surrogate loss, validating Thm 5.3.

### 6.3 COMPARISON WITH METHODS IN THE LITERATURE

Section 6.1 compares GSAM to SAM and vanilla training. In this subsection, we further compare GSAM against Entropy-SGD (Chaudhari et al., 2019) and Adaptive-SAM (ASAM) (Kwon et al., 2021), which are designed to improve generalization. Note that Entropy-SGD uses SGD in the inner Langevin iteration and can be combined with other base optimizers such as AdamW as the outer loop. For Entropy-SGD, we find the hyper-parameter "scope" from 0.0 and 0.9, and search for the inner-loop iteration number between 1 and 14. For ASAM, we search for $\rho$ between 1 and 7 ($10\times$ larger than in SAM) as recommended by the ASAM authors. Note that the only difference between ASAM and SAM is the derivation of the perturbation, so both can be combined with the proposed ascent step. As shown in Fig. 5, the proposed ascent step increases test accuracy when combined with both SAM and ASAM and outperforms Entropy-SGD and vanilla training.

### 6.4 ADDITIONAL STUDIES

**GSAM outperforms a weighted combination of the perturbed loss and surrogate gap** With an example in Fig. 2, we demonstrate that directly minimizing $f_p(w) + \lambda h(w)$ as discussed in Sec. 4.1 is sub-optimal because $\nabla h(w)$ could conflict with $\nabla f_p(w)$ and $\nabla f(w)$. We empirically validate this argument on ViT-B/32. We search for $\lambda$ between 0.0 and 0.5 with a step 0.1 and search for $\rho$ in the same grid as SAM and GSAM. We report the best accuracy of each method. Top-1 accuracy in Table 2 show the superior performance of GSAM, validating our analysis.

**$\min(f_p, h)$ vs. $\min(f, h)$** GSAM solves $\min(f_p, h)$ by descent in $\nabla f_p$, decomposing $\nabla f$ onto $\nabla f_p$, and an ascent step in the orthogonal direction to increase $f$ while keep $f_p$ intact. Alternatively, we can also optimize $\min(f, h)$ by descent in $\nabla f$, decomposing $\nabla f_p$ onto $\nabla f$, and a descent step in the orthogonal direction to decrease $f_p$ while keep $f$ intact. The two GSAM variations perform similarly (see Fig. 6, right). We choose $\min(f_p, h)$ mainly to make the minimal change to SAM.

**GSAM benefits transfer learning** Using weights trained on ImageNet-1k, we finetune models with SGD on downstream tasks including the CIFAR10/CIFAR100 (Krizhevsky et al., 2009), Oxford-

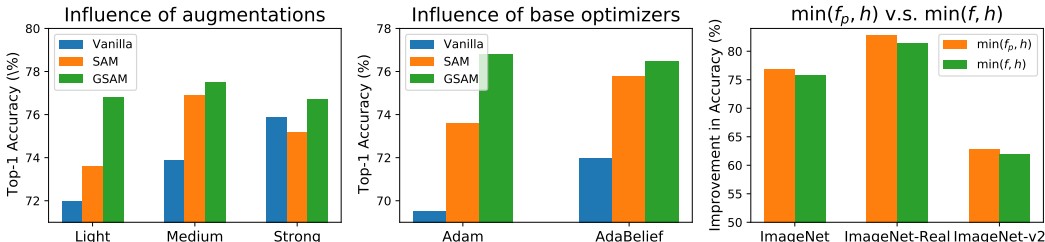

Figure 6: Top-1 accuracy of ViT-B/32 for the additional studies (Section 6.4). **Left**: from left to right are performances under different data augmentations (details in Appendix B.3) , where the vanilla method is trained for $2\times$ the epochs. **Middle**: performance with different base optimizers. **Right**: Comparison between $\min(f_p, h)$ and $\min(f, h)$.

flowers (Nilsback & Zisserman, 2008) and Oxford-IITPets (Parkhi et al., 2012). Results in Table 3 shows that GSAM leads to better transfer performance than vanilla training and SAM.

**GSAM remains effective under various data augmentations** We plot the top-1 accuracy of a ViT-B/32 model under various Mixup (Zhang et al., 2017b) augmentations in Fig. 6 (left subfigure). Under different augmentations, GSAM consistently outperforms SAM and vanilla training.

**GSAM is compatible with different base optimizers** GSAM is generic and applicable to various base optimizers. We compare vanilla training, SAM and GSAM using AdamW (Loshchilov & Hutter, 2017) and AdaBelief (Zhuang et al., 2020) with default hyper-parameters. Fig. 6 (middle subfigure) shows that GSAM performs the best, and SAM improves over vanilla training.

## 7 CONCLUSION

We propose the surrogate gap as an equivalent measure of sharpness which is easy to compute and feasible to optimize. We propose the GSAM method, which improves the generalization over SAM at negligible computation cost. We show the convergence and provably better generalization of GSAM compared to SAM, and validate the superior performance of GSAM on various models.

## ACKNOWLEDGEMENT

We would like to thank Xiangning Chen (UCLA) and Hossein Mobahi (Google) for discussions, Yi Tay (Google) for help with datasets, and Yeqing Li, Xianzhi Du, and Shawn Wang (Google) for help with TensorFlow implementation.

## ETHICS STATEMENT

This paper focuses on the development of optimization methodologies and can be applied to the training of different deep neural networks for a wide range of applications. Therefore, the ethical impact of our work would primarily be determined by the specific models that are trained using our new optimization strategy.

## REPRODUCIBILITY STATEMENT

We provide the detailed proof of theoretical results in Appendix A and provide the data pre-processing and hyper-parameter settings in Appendix B. Together with the references to existing works and public codebases, we believe the paper contains sufficient details to ensure reproducibility. We plan to release the models trained by using GSAM upon publication.

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

# A  PROOFS

## A.1  PROOF OF LEMMA. 3.1

Suppose $\rho$ is small, perform Taylor expansion around the local minima $w$, we have:

$$f(w + \delta) = f(w) + \nabla f(w)^\top \delta + \frac{1}{2}\delta^\top H \delta + O(||\delta||^3) \tag{10}$$

where $H$ is the Hessian, and is positive semidefinite at a local minima. At a local minima, $\nabla f(w) = 0$, hence we have

$$f(w + \delta) = f(w) + \frac{1}{2}\delta^\top H \delta + O(||\delta||^3) \tag{11}$$

and

$$f_p(w) = \max_{||\delta|| \le \rho} f(w + \delta) = f(w) + \frac{1}{2}\rho^2 \sigma_{max}(H) + O(||\delta||^3) \tag{12}$$

where $\sigma_{max}$ is the dominate eigenvalue (eigenvalue with the largest absolute value). Now consider two local minima $w_1$ and $w_2$ with dominate eigenvalue $\sigma_1$ and $\sigma_2$ respectively, we have

$$f_p(w_1) \approx f(w_1) + \frac{1}{2}\rho^2 \sigma_1 \qquad\qquad f_p(w_2) \approx f(w_2) + \frac{1}{2}\rho^2 \sigma_2$$

We have $f_p(w_1) > f_p(w_2) \not\Rightarrow \sigma_1 > \sigma_2$ and $\sigma_1 > \sigma_2 \not\Rightarrow f_p(w_1) > f_p(w_2)$ because the relation between $f(w_1)$ and $f(w_2)$ is undetermined. $\qquad\square$

## A.2  PROOF OF LEMMA. 3.2

Since $\rho$ is small, we can perform Taylor expansion around $w$,

$$\begin{aligned} h(w) &= f(w + \delta) - f(w) \\ &= \delta^\top \nabla f(w) + O(\rho^2) \\ &= \rho||\nabla f(w)||_2 + O(\rho^2) > 0 \end{aligned} \tag{13}$$

where the last line is because $\delta$ is approximated as $\delta = \rho\frac{\nabla f(w)}{||\nabla f(w)||_2 + \epsilon}$, hence has the same direction as $\nabla f(w)$. $\qquad\square$

## A.3  PROOF OF LEMMA. 3.3

Since $\rho$ is small, we can approximate $f(w)$ with a quadratic model around a local minima $w$:

$$f(w + \delta) = f(w) + \frac{1}{2}\delta^\top H \delta + O(\rho^3)$$

where $H$ is the Hessian at $w$, assumed to be positive semidefinite at local minima. Normalize $\delta$ such that $||\delta||_2 = \rho$, Hence we have:

$$h(w) = f_p(w) - f(w) = \max_{||\delta||_2 \le \rho} f(w + \delta) - f(w) = \frac{1}{2}\sigma_{max}\rho^2 + O(\rho^3) \tag{14}$$

where $\sigma_{max}$ is the dominate eigenvalue of the hessian $H$, and first order term is 0 because the gradient is 0 at local minima. Therefore, we have $\sigma_{max} \approx 2h(w)/\rho^2$. $\qquad\square$

## A.4  PROOF OF THM. 5.1

For simplicity we consider the base optimizer is SGD. For other optimizers such as Adam, we can derive similar results by applying standard proof techniques in the literature to our proof.

STEP 1: CONVERGENCE W.R.T FUNCTION $f_p(w)$

For simplicity of notation, we denote the update at step $t$ as

$$d_t = -\eta_t g_p^{(t)} + \eta_t \alpha g_\perp^{(t)} \tag{15}$$

By $L-$smoothness of $f$ and the definition of $f_p(w_t) = f(w_t^{adv})$, and definition of $d_t = w_{t+1} - w_t$ and $w_t^{adv} = w_t + \delta_t$ we have

$$f_p(w_{t+1}) = f(w_{t+1}^{adv}) \leq f(w_t^{adv}) + \langle \nabla f(w_t^{adv}), w_{t+1}^{adv} - w_t^{adv} \rangle + \frac{L}{2}\left|\left|w_{t+1}^{adv} - w_t^{adv}\right|\right|^2 \tag{16}$$

$$= f(w_t^{adv}) + \langle \nabla f(w_t^{adv}), w_{t+1} + \delta_{t+1} - w_t - \delta_t \rangle$$
$$+ \frac{L}{2}\left|\left|w_{t+1} + \delta_{t+1} - w_t - \delta_t\right|\right|^2 \tag{17}$$

$$\leq f(w_t^{adv}) + \langle \nabla f(w_t^{adv}), d_t \rangle + L\left|\left|d_t\right|\right|^2 \tag{18}$$

$$+ \langle \nabla f(w_t^{adv}), \delta_{t+1} - \delta_t \rangle + L\left|\left|\delta_{t+1} - \delta_t\right|\right|^2 \tag{19}$$

STEP 1.0: BOUND EQ. 18

We first bound Eq. 18. Take expectation conditioned on observation up to step $t$ (for simplicity of notation, we use $\mathbb{E}$ short for $\mathbb{E}_x$ to denote expectation over all possible data points) conditioned on observations up to step $t$, also by definition of $d_t$, we have

$$\mathbb{E}f_p(w_{t+1}) - f_p(w_t) \leq -\eta_t \langle \nabla f_p(w_t), \mathbb{E}g_p^{(t)} \rangle + \alpha \eta_t \langle \nabla f_p(w_t), \mathbb{E}g_\perp^{(t)} \rangle$$
$$+ L\eta_t^2 \mathbb{E}\left|\left| -g_p^{(t)} + \alpha g_\perp^{(t)}\right|\right|_2^2 \tag{20}$$

$$\leq -\eta_t \mathbb{E}\left|\left|\nabla f_p(w_t)\right|\right|_2^2 + 0 + (\alpha + 1)^2 G^2 \eta_t^2 \tag{21}$$

$$\left(\text{Since } \mathbb{E}g_\perp^{(t)} \text{ is orthogonal to } \nabla f_p(w_t) \text{ by construction,}\right.$$

$$\left. ||g^{(t)}|| \leq G \text{ by assumption}\right)$$

STEP 1.1: BOUND EQ. 19

By definition of $\delta_t$, we have

$$\delta_t = \rho_t \frac{g^{(t)}}{||g^{(t)}|| + \epsilon} \tag{22}$$

$$\delta_{t+1} = \rho_{t+1} \frac{g^{(t+1)}}{||g^{(t+1)}|| + \epsilon} \tag{23}$$

where $g^{(t)}$ is the gradient of $f$ at $w_t$ evaluated with a noisy data sample. When learning rate $\eta_t$ is small, the update in weight $d_t$ is small, and expected gradient is

$$\nabla f(w_{t+1}) = \nabla f(w_t + d_t) = \nabla f(w_t) + Hd_t + O(||d_t||^2) \tag{24}$$

where $H$ is the Hessian at $w_t$. Therefore, we have

$$\mathbb{E}\langle \nabla f(w_t^{adv}), \delta_{t+1} - \delta_t \rangle = \langle \nabla f(w_t^{adv}), \rho_t \mathbb{E}\frac{g^{(t)}}{||g^{(t)}|| + \epsilon} - \rho_{t+1}\mathbb{E}\frac{g^{(t+1)}}{||g^{(t+1)}|| + \epsilon} \rangle \tag{25}$$

$$\leq ||\nabla f(w_t^{adv})||\rho_t \left|\left|\mathbb{E}\frac{g^{(t)}}{||g^{(t)}|| + \epsilon} - \mathbb{E}\frac{g^{(t+1)}}{||g^{(t+1)}|| + \epsilon}\right|\right| \tag{26}$$

$$\leq ||\nabla f(w_t^{adv})||\rho_t \phi_t \tag{27}$$

where the first inequality is due to (1) $\rho_t$ is monotonically decreasing with $t$, and (2) triangle inequality that $\langle a, b \rangle \leq ||a|| \cdot ||b||$. $\phi_t$ is the angle between the unit vector in the direction of $\nabla f(w_t)$

and $\nabla f(w_{t+1})$. The second inequality comes from that (1) $\left|\left|\frac{g}{||g||+\epsilon}\right|\right| < 1$ strictly, so we can replace $\delta_t$ in Eq. 25 with a unit vector in corresponding directions multiplied by $\rho_t$ and get the upper bound, (2) the norm of difference in unit vectors can be upper bounded by the arc length on a unit circle.

When learning rate $\eta_t$ and update stepsize $d_t$ is small, $\phi_t$ is also small. Using the limit that

$$\tan x = x + O(x^2), \quad \sin x = x + O(x^2), \quad x \to 0$$

We have:

$$\tan \phi_t = \frac{||\nabla f(w_{t+1}) - \nabla f(w_t)||}{||\nabla f(w_t)||} + O(\phi_t^2) \tag{28}$$

$$= \frac{||Hd_t + O(||d_t||^2)||}{||\nabla f(w_t)||} + O(\phi_t^2) \tag{29}$$

$$\leq \eta_t L(1 + \alpha) \tag{30}$$

where the last inequality is due to (1) max eigenvalue of $H$ is upper bounded by $L$ because $f$ is $L$−smooth, (2) $||d_t|| = ||\eta_t(g_\parallel + \alpha g_\perp)||$ and $\mathbb{E}g_t = \nabla f(w_t)$.

Plug into Eq. 27, also note that the perturbation amplitude $\rho_t$ is small so $w_t$ is close to $w_t^{adv}$, then we have

$$\mathbb{E}\langle \nabla f(w_t^{adv}), \delta_{t+1} - \delta_t \rangle \leq L(1 + \alpha)G\rho_t\eta_t \tag{31}$$

Similarly, we have

$$\mathbb{E}\left|\left|\delta_{t+1} - \delta_t\right|\right|^2 \leq \rho_t^2 \mathbb{E}\left|\left|\frac{g^{(t)}}{||g^{(t)}|| + \epsilon} - \frac{g^{(t+1)}}{||g^{(t+1)}|| + \epsilon}\right|\right|^2 \tag{32}$$

$$\leq \rho_t^2 \phi_t^2 \tag{33}$$

$$\leq \rho_t^2 \eta_t^2 L^2 (1 + \alpha)^2 \tag{34}$$

STEP 1.2: TOTAL BOUND

Reuse results from Eq. 21 (replace $L_p$ with $2L$) and plug into Eq. 18, and plug Eq. 31 and Eq. 34 into Eq. 19, we have

$$\mathbb{E}f_p(w_{t+1}) - f_p(w_t) \leq -\eta_t \mathbb{E}\left|\left|\nabla f_p(w_t)\right|\right|_2^2 + \frac{2L(\alpha + 1)^2}{2}G^2\eta_t^2$$
$$+ L(1 + \alpha)G\rho_t\eta_t + \frac{2L^3(1 + \alpha)^2}{2}\eta_t^2\rho_t^2 \tag{35}$$

Perform telescope sum, we have

$$\mathbb{E}f_p(w_T) - f_p(w_0) \leq -\sum_{t=1}^{T}\eta_t \mathbb{E}||\nabla f_p(w_t)||^2 + \left[L(1 + \alpha)^2 G^2 \eta_0^2 + L(1 + \alpha)G\rho_0\eta_0\right]\sum_{t=1}^{T}\frac{1}{t}$$
$$+ L^3(1 + \alpha)^2\eta_0^2\rho_0^2 \sum_{t=1}^{T}\frac{1}{t^2} \tag{36}$$

Hence

$$\eta_T \sum_{t=1}^{T}\mathbb{E}||\nabla f_p(w_t)||^2 \leq \sum_{t=1}^{T}\eta_t \mathbb{E}||\nabla f_p(w_t)||^2 \leq f_p(w_0) - \mathbb{E}f_p(w_T) + D\log T + \frac{\pi^2 E}{6} \tag{37}$$

where

$$D = L(1 + \alpha)^2 G^2 \eta_0^2 + L(1 + \alpha)G\rho_0\eta_0, \quad E = L^3(1 + \alpha)^2\eta_0^2\rho_0^2 \tag{38}$$

Note that $\eta_T = \frac{\eta_0}{\sqrt{T}}$, we have

$$\frac{1}{T}\sum_{t=1}^{T}\mathbb{E}||\nabla f_p(w_t)||^2 \leq \frac{f_p(w_0) - f_{min} + \pi^2 E/6}{\eta_0}\frac{1}{\sqrt{T}} + \frac{D}{\eta_0}\frac{\log T}{\sqrt{T}} \tag{39}$$

which implies that GSAM enables $f_p$ to converge at a rate of $O(\log T/\sqrt{T})$, and all the constants here are well-bounded.

STEP 2: CONVERGENCE W.R.T. FUNCTION $f(w)$

We prove the risk for $f(w)$ convergences for non-convex stochastic optimization case using SGD. Denote the update at step $t$ as

$$d_t = -\eta_t g_p^{(t)} + \alpha \eta_t g_\perp^{(t)} \tag{40}$$

By smoothness of $f$, we have

$$f(w_{t+1}) \leq f(w_t) + \langle \nabla f(w_t), d_t \rangle + \frac{L}{2} \left\| d_t \right\|_2^2 \tag{41}$$

$$= f(w_t) + \langle \nabla f(w_t), -\eta_t g_p^{(t)} + \alpha \eta_t g_\perp^{(t)} \rangle + \frac{L}{2} \left\| d_t \right\|_2^2 \tag{42}$$

For simplicity, we introduce a scalar $\beta_t$ such that

$$\nabla f_\|(w_t) = \beta_t \nabla f_p(w_t) \tag{43}$$

where $\nabla f_\|(w_t)$ is the projection of $\nabla f(w_t)$ onto $\nabla f_p(w_t)$. When perturbation amplitude $\rho$ is small, we expect $\beta_t$ to be very close to 1.

Take expectation conditioned on observations up to step $t$ for both sides of Eq. 42, we have:

$$\mathbb{E}f(w_{t+1}) \leq f(w_t) + \left\langle \nabla f(w_t), -\frac{\eta_t}{\beta_t}\left(\nabla f(w_t) - \nabla f_\perp(w_t)\right) + \alpha \eta_t \mathbb{E} g_\perp^{(t)} \right\rangle + \frac{L}{2} \mathbb{E} \left\| d_t \right\|_2^2 \tag{44}$$

$$= f(w_t) - \frac{\eta_t}{\beta_t} \left\| \nabla f(w_t) \right\|_2^2 + \left(\frac{1}{\beta_t} + \alpha\right)\eta_t \left\langle \nabla f(w_t), \nabla f_\perp(w_t) \right\rangle + \frac{L}{2} \mathbb{E} \left\| d_t \right\|_2^2 \tag{45}$$

$$= f(w_t) - \frac{\eta_t}{\beta_t} \left\| \nabla f(w_t) \right\|_2^2 + \left(\frac{1}{\beta_t} + \alpha\right)\eta_t \left\langle \nabla f(w_t), \nabla f(w_t) \sin \theta_t \right\rangle + \frac{L}{2} \mathbb{E} \left\| d_t \right\|_2^2 \tag{46}$$

$$\left(\theta_t \text{ is the angle between } \nabla f_p(w_t) \text{ and } \nabla f(w_t)\right)$$

$$= f(w_t) - \frac{\eta_t}{\beta_t} \left\| \nabla f(w_t) \right\|_2^2 + \left(\frac{1}{\beta_t} + \alpha\right)\eta_t \left\| \nabla f(w_t) \right\|_2^2 (|\tan \theta_t| + O(\theta_t^2)) + \frac{L}{2} \mathbb{E} \left\| d_t \right\|_2^2 \tag{47}$$

$$\left(\sin x = x + O(x^2), \tan x = x + O(x^2) \text{ when } x \to 0.\right)$$

Also note when perturbation amplitude $\rho_t$ is small, we have

$$\nabla f_p(w_t) = \nabla f(w_t + \delta_t) = \nabla f(w_t) + \frac{\rho_t}{||\nabla f(w_t)||_2 + \epsilon} H(w_t)\nabla f(w_t) + O(\rho_t^2) \tag{48}$$

where $\delta_t = \rho_t \frac{\nabla f(w_t)}{||\nabla f(w_t)||_2}$ by definition, $H(w_t)$ is the Hessian. Hence we have

$$|\tan \theta_t| \leq \frac{||\nabla f_p(w_t) - \nabla f(w_t)||}{||\nabla f(w_t)||} \leq \frac{\rho_t L}{||\nabla f(w_t)||} \tag{49}$$

where $L$ is the Lipschitz constant of $f$, and $L-$smoothness of $f$ indicates the maximum absolute eigenvalue of $H$ is upper bounded by $L$. Plug Eq. 49 into Eq. 47, we have

$$\mathbb{E}f(w_{t+1}) \leq f(w_t) - \frac{\eta_t}{\beta_t} \left\| \nabla f(w_t) \right\|_2^2 + \left(\frac{1}{\beta_t} + \alpha\right)\eta_t \left\| \nabla f(w_t) \right\|_2^2 |\tan \theta_t| + \frac{L}{2} \mathbb{E} \left\| d_t \right\|_2^2 \tag{50}$$

$$\leq f(w_t) - \frac{\eta_t}{\beta_t} \left\| \nabla f(w_t) \right\|_2^2 + \left(\frac{1}{\beta_t} + \alpha\right) L \rho_t \eta_t \left\| \nabla f(w_t) \right\|_2 + \frac{L}{2} \mathbb{E} \left\| d_t \right\|_2^2 \tag{51}$$

$$\leq f(w_t) - \frac{\eta_t}{\beta_t} \left\| \nabla f(w_t) \right\|_2^2 + \left(\frac{1}{\beta_t} + \alpha\right) L \rho_t \eta_t G + \frac{L}{2} \mathbb{E} \left\| d_t \right\|_2^2 \tag{52}$$

$$\left(\text{Assume gradient has bounded norm } G.\right) \tag{53}$$

$$\leq f(w_t) - \frac{\eta_t}{\beta_{max}} \left\| \nabla f(w_t) \right\|_2^2 + \left(\frac{1}{\beta_{min}} + \alpha\right) L \rho_t \eta_t G + \frac{L}{2} \mathbb{E}(\alpha + 1)^2 G^2 \eta_t^2 \tag{54}$$

$$\left(\beta_t \text{ is close to 1 assuming } \rho \text{ is small},\right.$$

$$\left.\text{hence it's natural to assume } 0 < \beta_{min} \leq \beta_t \leq \beta_{max}\right)$$

Re-arranging above formula, we have

$$\frac{\eta_t}{\beta_{max}}\left|\left|\nabla f(w_t)\right|\right|_2^2 \le f(w_t) - \mathbb{E}f(w_{t+1}) + \left(\frac{1}{\beta_{min}} + \alpha\right)LG\eta_t\rho_t + \frac{L}{2}(\alpha+1)^2 G^2\eta_t^2 \quad (55)$$

perform telescope sum and taking expectations on each step, we have

$$\frac{1}{\beta_{max}}\sum_{t=1}^{T}\eta_t\left|\left|\nabla f(w_t)\right|\right|_2^2 \le f(w_0) - \mathbb{E}f(w_T) + \left(\frac{1}{\beta_{min}} + \alpha\right)LG\sum_{t=1}^{T}\eta_t\rho_t + \frac{L}{2}(\alpha+1)^2 G^2\sum_{t=1}^{T}\eta_t^2$$
$$(56)$$

Take the schedule to be $\eta_t = \frac{\eta_0}{\sqrt{t}}$ and $\rho_t = \frac{\rho_0}{\sqrt{t}}$, then we have

$$\frac{\eta_0}{\beta_{max}}\frac{1}{\sqrt{T}}\sum_{t=1}^{T}\left|\left|\nabla f(w_t)\right|\right|_2^2 \le LHS \quad (57)$$

$$\le RHS \quad (58)$$

$$\le f(w_0) - f_{min} + \left(\frac{1}{\beta_{min}} + \alpha\right)LG\eta_0\rho_0\sum_{t=1}^{T}\frac{1}{t} + \frac{L}{2}(\alpha+1)^2 G^2\eta_0^2\sum_{t=1}^{T}\frac{1}{t}$$
$$(59)$$

$$\le f(w_0) - f_{min} + \left(\frac{1}{\beta_{min}} + \alpha\right)LG\eta_0\rho_0(1 + \log T)$$
$$+ \frac{L}{2}(\alpha+1)^2 G^2\eta_0^2(1 + \log T) \quad (60)$$

Hence

$$\frac{1}{T}\sum_{t=1}^{T}\left|\left|\nabla f(w_t)\right|\right|_2^2 \le \frac{C_3}{\sqrt{T}} + C_4\frac{\log T}{\sqrt{T}} \quad (61)$$

where $C_1, C_4$ are some constants. This implies the convergence rate w.r.t $f(w)$ is $O(\log T/\sqrt{T})$.

STEP 3: CONVERGENCE W.R.T. SURROGATE GAP $h(w)$

Note that we have proved convergence for $f_p(w)$ in step 1, and convergence for $f(w)$ in step 3. Also note that

$$\left|\left|\nabla h(w_t)\right|\right|_2^2 = \left|\left|\nabla f_p(w_t) - \nabla f(w_t)\right|\right|_2^2 \le 2\left|\left|\nabla f_p(w_t)\right|\right|_2^2 + 2\left|\left|\nabla f(w_t)\right|\right|_2^2 \quad (62)$$

Hence

$$\frac{1}{T}\sum_{t=1}^{T}\left|\left|\nabla h(w_t)\right|\right|_2^2 \le \frac{2}{T}\sum_{t=1}^{T}\left|\left|\nabla f_p(w_t)\right|\right|_2^2 + \frac{2}{T}\sum_{t=1}^{T}\left|\left|\nabla f(w_t)\right|\right|_2^2 \quad (63)$$

also converges at rate $O(\log T/\sqrt{T})$ because each item in the RHS converges at rate $O(\log T\sqrt{T})$.
$\square$

## A.5 PROOF OF COROLLARY. 5.2.1

Using the results from Thm. 5.2, with probability at least $1 - a$, we have

$$\mathbb{E}_{w\sim\mathcal{Q}}\mathbb{E}_x f(w, x) \le \mathbb{E}_{w\sim\mathcal{Q}}\widehat{f}(w) + 4\sqrt{\frac{KL(\mathcal{Q}||\mathcal{P}) + \log\frac{2m}{a}}{m}} \quad (64)$$

Assume $\delta \sim \mathcal{N}(0, b^2 I_k)$ where $k$ is the dimension of model parameters, hence $\delta^2$ (element-wise square) follows a a Chi-square distribution. By Lemma.1 in Laurent & Massart (2000), we have

$$\mathbb{P}\left(||\delta||_2^2 - kb^2 \ge 2b^2\sqrt{kt} + 2tb^2\right) \le exp(-t) \quad (65)$$

hence with probability at least $1 - 1/\sqrt{n}$, we have

$$||\delta||_2^2 \le b^2\left(2\log\sqrt{n} + k + 2\sqrt{k\log\sqrt{n}}\right) \le 2b^2 k\left(1 + \sqrt{\frac{\log\sqrt{n}}{k}}\right)^2 \le \rho^2 \quad (66)$$

Therefore, with probability at least $1 - 1/\sqrt{n} = 1 - exp\left(-\left(\frac{\rho}{\sqrt{2b}} - \sqrt{k}\right)^2\right)$

$$\mathbb{E}_\delta \widehat{f}(w+\delta) \leq \max_{||\delta||_2 \leq \rho} \widehat{f}(w+\delta) \tag{67}$$

Combine Eq. 65 and Eq. 67, subtract the same constant $C$ on both sides, and under the same assumption as in (Foret et al., 2020) that $\mathbb{E}_{w \sim \mathcal{Q}} \mathbb{E}_x f(w,x) \leq \mathbb{E}_{\delta \sim \mathcal{N}(0,b^2 I^k)} \mathbb{E}_{w \sim \mathcal{Q}} \mathbb{E}_x f(w+\delta, x)$ we finish the proof. $\qquad \square$

### A.6  PROOF OF THM. 5.3

STEP 1: A SUFFICIENT CONDITION THAT THE LOSS GAP IS EXPECTED TO DECREASE FOR EACH STEP

Take Taylor expansion, then the expected change of loss gap caused by descent step is

$$\mathbb{E}\langle \nabla f_p(w_t) - \nabla f(w_t), -\eta_t \nabla f_p(w_t)\rangle \tag{68}$$
$$\left(where\ \mathbb{E}g_\perp = \nabla f_\perp(w_t)\right)$$
$$= \eta_t \left[ -||\nabla f_p(w_t)||_2^2 + ||\nabla f_p(w_t)||_2 ||\nabla f(w_t)||_2 \cos\theta_t \right] \tag{69}$$

where $\theta_t$ is the angle between vector $\nabla f_p(w_t)$ and $\nabla f(w_t)$.
The expected change of loss gap caused by ascent step is

$$\mathbb{E}\langle \nabla f_p(w_t) - \nabla f(w_t), \alpha\eta_t \nabla f_\perp(w_t)\rangle = -\alpha\eta_t ||\nabla f_\perp(w_t)||_2^2 < 0 \tag{70}$$

Above results demonstrate that ascent step decreases the loss gap, while descent step might increase the loss gap. A sufficient (but not necessary) condition for $\mathbb{E}\langle \nabla h(w_t), dt\rangle \leq 0$ requires $\alpha$ to be large or $||\nabla f(w_t)||_2 \cos\theta_t \leq ||\nabla f_p(w_t)||$. In practice, the perturbation amplitude $\rho$ is small and we can assume $\theta_t$ is close to 0 and $||\nabla f_p(w_t)||$ is close to $||\nabla f(w_t)||$, we can also set the parameter $\alpha$ to be large in order to decrease the loss gap.

STEP 2: UPPER AND LOWER BOUND OF DECREASE IN LOSS GAP (BY THE ASCENT STEP IN ORTHOGONAL GRADIENT DIRECTION) COMPARED TO SAM.

Next we give an estimate of the decrease in $\widehat{h}$ caused by our ascent step. We refer to Eq. 69 and Eq. 70 to analyze the change in loss gap caused by the descent and ascent (orthogonally) respectively. It can be seen that gradient descent step might not decrease loss gap, in fact they often increase loss gap in practice; while the ascent step is guaranteed to decrease the loss gap.

The decrease in loss gap is:

$$\Delta \widehat{h}_t = -\langle \nabla \widehat{f}_p(w_t) - \nabla \widehat{f}(w_t), \alpha\eta_t \nabla \widehat{f}_\perp(w_t)\rangle = \alpha\eta_t ||\nabla \widehat{f}_\perp(w_t)||_2^2 \tag{71}$$
$$= \alpha\eta_t ||\nabla \widehat{f}(w_t)||_2^2 |\tan\theta_t|^2 \tag{72}$$

$$\sum_{t=1}^{T} \Delta \widehat{h}_t \leq \sum_{t=1}^{T} \alpha L^2 \eta_t \rho_t^2 \tag{73}$$
$$\left(\text{By Eq. 49}\right) \tag{74}$$
$$\leq \sum_{t=1}^{T} \alpha L^2 \eta_0 \rho_0^2 \frac{1}{t^{3/2}} \tag{75}$$
$$\leq 2.7 \alpha L^2 \eta_0 \rho_0^2 \tag{76}$$

Hence we derive an upper bound for $\sum_{t=1}^{T} \Delta \widehat{h}_t$.

Next we derive a lower bound for $\sum_{t=1}^{T} \Delta \widehat{h}_t$ Note that when $\rho_t$ is small, by Taylor expansion

$$\nabla \widehat{f}_p(w_t) = \nabla \widehat{f}(w_t + \delta_t) = \nabla \widehat{f}(w_t) + \frac{\rho_t}{||\nabla \widehat{f}(w_t)||} \widehat{H}(w_t) \nabla \widehat{f}(w_t) + O(\rho_t^2) \tag{77}$$

where $\widehat{H}(w_t)$ is the Hessian evaluated on training samples. Also when $\rho_t$ is small, the angle $\theta_t$ between $\nabla \widehat{f}_p(w_t)$ and $\nabla \widehat{f}(w_t)$ is small, by the limit that

$$\tan x = x + O(x^2), x \to 0$$
$$\sin x = x + O(x^2), x \to 0$$

We have

$$|\tan \theta_t| = |\sin \theta_t| + O(\theta_t^2) = |\theta_t| + O(\theta_t^2)$$

Omitting high order term, we have

$$|\tan \theta_t| \approx |\theta_t| = \frac{||\nabla \widehat{f}_p(w_t) - \nabla \widehat{f}(w_t)||}{||\widehat{f}(w_t)||} = \frac{||\rho_t \widehat{H}(w_t) + O(\rho_t^2)||}{||\nabla \widehat{f}(w_t)||} \geq \frac{\rho_t |\sigma|_{min}}{G} \tag{78}$$

where $G$ is the upper-bound on norm of gradient, $|\sigma|_{min}$ is the minimum absolute eigenvalue of the Hessian. The intuition is that as perturbation amplitude decreases, the angle $\theta_t$ decreases at a similar rate, though the scale constant might be different. Hence we have

$$\sum_{t=1}^{T} \Delta \widehat{h}_t = \sum_{t=1}^{T} \alpha \eta_t ||\nabla \widehat{f}(w_t)||_2^2 |\tan \theta_t|^2 + O(\theta_t^4) \tag{79}$$

$$\geq \sum_{t=1}^{T} \alpha \eta_t c^2 \left( \frac{\rho_t |\sigma|_{min}}{G} \right)^2 \tag{80}$$

$$= \frac{\alpha c^2 \rho_0^2 \eta_0 |\sigma|_{min}^2}{G^2} \sum_{t=1}^{T} \frac{1}{t^{3/2}} \tag{81}$$

$$\geq \frac{\alpha c^2 \rho_0^2 \eta_0 |\sigma|_{min}^2}{G^2} \tag{82}$$

where $c^2$ is the lower bound of $||\nabla \widehat{f}||^2$ (e.g. due to noise in data and gradient observation). Results above indicate that the decrease in loss gap caused by the ascent step is non-trivial, hence our proposed method efficiently improves generalization compared with SAM. □

## A.7 DISCUSSION ON COROLLARY 5.2.1

The comment "'The corollary gives a bound on the risk in terms of the perturbed training loss if one removes $C$ from both sides'" is correct. But there is a misunderstanding in the statement "'the perturbed training loss is small then the model has a small risk'": it's only true when $\rho_{train}$ for training equals its real value $\rho_{true}$ determined by the data distribution; in practice, we never know $\rho_{true}$. In the following we show that the minimization of both $h$ and $f_p$ is better than simply minimizing $f_p$ **when $\rho_{true} \neq \rho_{train}$.**

1. First, we re-write the conclusion of Corollary 5.2.1 as

$$\mathbb{E}_w \mathbb{E}_x f(w, x) \leq f_p + R = C + \widehat{h} + R = C + \rho^2 \sigma/2 + R + O(\rho^3)$$
$$\text{with probability } (1 - a)[1 - e^{-(\frac{\rho}{\sqrt{2}b} - \sqrt{k})^2}]$$

where $R$ is the regularization term, $C$ is the training loss, $\sigma$ is the dominant eigenvalue of Hessian. As in lemma 3.3, we perform Taylor-expansion and can ignore the high-order term $O(\rho^3)$. We focus on

$$f_p = C + \widehat{h} = C + \rho^2 \sigma/2$$

2. **When $\rho_{true} \neq \rho_{train}$, minimizing $h$ achieves a lower risk than only minimizing $f_p$.** (1) Note that after training, $C$ **(training loss) is fixed, but $h$ could vary with** $\rho$ (e.g. when training on dataset A and testing on an unrelated dataset B, the training loss remains unchanged, but the risk would be huge and a large $\rho$ is required for a valid bound). (2) With an example, we show a low $f_p$ is insufficient for generalization, and a low $\sigma$ is necessary:

A Suppose we use $\rho_{train}$ for training, and consider two solutions with $C_1, \sigma_1$ (SAM) and $C_2, \sigma_2$ (GSAM). Suppose they have the same $f_p$ during training for some $\rho_{train}$, so

$$f_{p1} = C_1 + \sigma_1/2 \times \rho_{train}^2 = C_2 + \sigma_2/2 \times \rho_{train}^2 = f_{p2}$$

Suppose $C_1 < C_2$ so $\sigma_1 > \sigma_2$.

B When $\rho_{true} > \rho_{train}$, we have

$\texttt{risk\_bound\_1} = C_1 + \sigma_1/2 \times \rho_{true}^2 + R > \texttt{risk\_bound\_2} = C_2 + \sigma_2/2 \times \rho_{true}^2 + R$

This implies that a **small $\sigma$ helps generalization**, but only a low $f_{p1}$ (caused by a low $C_1$ and high $\sigma_1$) is insufficient for a good generalization.

C Note that $\rho_{train}$ is fixed during training, so minimizing $h_{train}$ during training is equivalently minimizing $\sigma$ by Lemma 3.3

3. **Why we are often unlucky to have $\rho_{true} > \rho_{train}$** (1) First, the test sets are almost surely **outside** the convex hull of the training set because "'interpolation almost surely never occurs in high-dimensional ($> 100$) cases'" Balestriero et al. (2021). As a result, the variability of (train + test) sets is almost surely larger than the variability of (train) set. Since $\rho$ increases with data variability (see point 4 below), we have $\rho_{true} > \rho_{train\_set}$ almost surely. (2) Second, we don't know the value of $\rho_{true}$ and can only guess it. In practice, we often guess a small value because training often diverges with large $\rho$ (as observed in Foret et al. (2020); Chen et al. (2021)).

4. **Why $\rho$ increases with data variability.** In Corollary 5.2.1, we assume weight perturbation $\delta \sim \mathcal{N}(0, b^2 I^k)$. The meaning of $b$ is the following. If we can randomly sample a fixed number of samples from the underlying distribution, then training the model from scratch (with a fixed seed for random initialization) gives rise to a set of weights. Repeating this process, we get many sets of weights, and their standard deviation is $b$. Since the number of training samples is limited and fixed, the more variability in data, the more variability in weights, and the larger $b$. Note that Corollary stated that the bound holds with probability proportional to $[1 - e^{-(\frac{\rho}{\sqrt{2}b} - \sqrt{k})^2}]$. In order for the result to hold with a fixed probability, $\rho$ must stay proportional to $b$, hence $\rho$ also increases with the variability of data.

Table 4: Hyper-parameters to reproduce experimental results

| Model | $\rho_{max}$ | $\rho_{min}$ | $\alpha$ | $lr_{max}$ | $lr_{min}$ | Weight Decay | Base Optimizer | Epochs | Warmup Steps | LR schedule |
|---|---|---|---|---|---|---|---|---|---|---|
| ResNet50 | 0.04 | 0.02 | 0.01 | 1.6 | 1.6e-2 | 0.3 | SGD | 90 | 5k | Linear |
| ResNet101 | 0.04 | 0.02 | 0.01 | 1.6 | 1.6e-2 | 0.3 | SGD | 90 | 5k | Linear |
| ResNet512 | 0.04 | 0.02 | 0.005 | 1.6 | 1.6e-2 | 0.3 | SGD | 90 | 5k | Linear |
| ViT-S/32 | 0.6 | 0.0 | 0.4 | 3e-3 | 3e-5 | 0.3 | AdamW | 300 | 10k | Linear |
| ViT-S/16 | 0.6 | 0.0 | 1.0 | 3e-3 | 3e-5 | 0.3 | AdamW | 300 | 10k | Linear |
| ViT-B/32 | 0.6 | 0.1 | 0.6 | 3e-3 | 3e-5 | 0.3 | AdamW | 300 | 10k | Linear |
| ViT-B/16 | 0.6 | 0.2 | 0.4 | 3e-3 | 3e-5 | 0.3 | AdamW | 300 | 10k | Linear |
| Mixer-S/32 | 0.5 | 0.0 | 0.2 | 3e-3 | 3e-5 | 0.3 | AdamW | 300 | 10k | Linear |
| Mixer-S/16 | 0.5 | 0.0 | 0.6 | 3e-3 | 3e-5 | 0.3 | AdamW | 300 | 10k | Linear |
| Mixer-S/8 | 0.5 | 0.1 | 0.1 | 3e-3 | 3e-5 | 0.3 | AdamW | 300 | 10k | Linear |
| Mixer-B/32 | 0.7 | 0.2 | 0.05 | 3e-3 | 3e-5 | 0.3 | AdamW | 300 | 10k | Linear |
| Mixer-B/16 | 0.5 | 0.2 | 0.01 | 3e-3 | 3e-5 | 0.3 | AdamW | 300 | 10k | Linear |

## B  EXPERIMENTAL DETAILS

### B.1  TRAINING DETAILS

For ViT and Mixer, we search the learning rate in {1e-3, 3e-3, 1e-2, 3e-3}, and search weight decay in {0.003, 0.03, 0.3}. For ResNet, we search the learning rate in {1.6, 0.16, 0.016}, and search the weight decay in {0.001, 0.01,0.1}. For ViT and Mixer, we use the AdamW optimizer with $\beta_1 = 0.9, \beta_2 = 0.999$; for ResNet we use SGD with momentum= 0.9. We train ResNets for 90 epochs, and train ViTs and Mixers for 300 epochs following the settings in (Chen et al., 2021) and (Dosovitskiy et al., 2020). Considering that SAM and GSAM uses twice the computation of vanilla training for each step, for vanilla training we try $2\times$ longer training, and does not find significant improvement as in Table. 5.

We first search the optimal learning rate and weight decay for vanilla training, and keep these two hyper-parameters fixed for SAM and GSAM. For ViT and Mixer, we search $\rho$ in {0.1, 0.2, 0.3, 0.4, 0.5, 0.6} for SAM and GSAM; for ResNet, we search $\rho$ from 0.01 to 0.05 with a stepsize 0.01. For ASAM, we amplify $\rho$ by $10\times$ compared to SAM, as recommended by Kwon et al. (2021). For GSAM, we search $\alpha$ in {0.1, 0.2, 0.3} throughout the paper. We report the best configuration of each individual model in Table. 4.

### B.2  TRANSFER LEARNING EXPERIMENTS

Using weights trained on ImageNet-1k, we finetune models with SGD on downstream tasks including the CIFAR10/CIFAR100 (Krizhevsky et al., 2009), Oxford-flowers (Nilsback & Zisserman, 2008) and Oxford-IITPets (Parkhi et al., 2012). For all experiments, we use the SGD optimizer with no weight decay under a linear learning rate schedule and gradient clipping with global norm 1. We search the maximum learning rate in {0.001, 0.003, 0.01, 0.03}. On Cifar datasets, we train models for 10k steps with a warmup step of 500; on Oxford datasets, we train models for 500 steps with a wamup step of 100.

### B.3  EXPERIMENTAL SETUP WITH ABLATION STUDIES ON DATA AUGMENTATION

We follow the settings in (Tolstikhin et al., 2021) to perform ablation studies on data augmentation. In the left subfigure of Fig. 6, "Light" refers to Inception-style data augmentation with random flip and crop of images, "Medium" refers to the mixup augmentation with probability 0.2 and RandAug magnitude 10; "Strong" refers to the mixup augmentation with probability 0.2 and RandAug magnitude 15.

## C  ABLATION STUDIES AND DISCUSSIONS

### C.1  INFLUENCE OF $\rho$ AND $\alpha$

We plot the performance of a ViT-B/32 model varying with $\rho$ (Fig. 7a) and $\alpha$ (Fig. 7b). We empirically validate that fine-tuning $\rho$ in SAM can not achieve comparable performance with GSAM, as

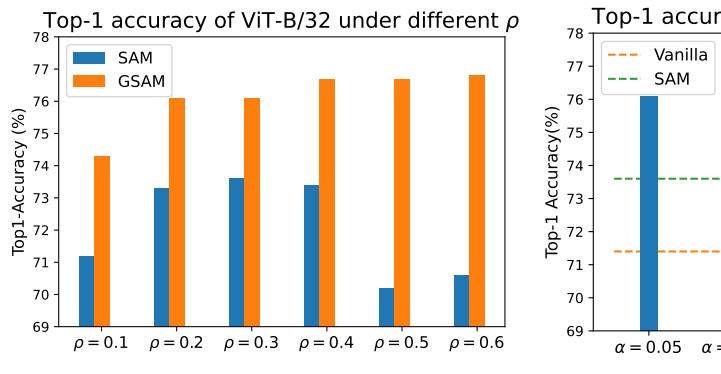

(a) Performance of SAM and GSAM under different $\rho$.  (b) Performance of GSAM under different $\alpha$

Figure 7: Performance of GSAM varying with $\rho$ and $\alpha$.

Table 5: Top-1 accuracy of ViT-B/32 on ImageNet with Inception-style data augmentation. For vanilla training we report results for training 300 epochs and 600 epochs, for GSAM we report the results for 300 epochs.

| Method | Epochs | ImageNet | ImageNet-Real | ImageNet-v2 | ImageNet-R |
|--------|--------|----------|---------------|-------------|------------|
| Vanilla | 300 | 71.4 | 77.5 | 57.5 | 23.4 |
|         | 600 | 72.0 | 78.2 | 57.9 | 23.6 |
| GSAM | 300 | **76.8** | **82.7** | **63.0** | **25.1** |

shown in Fig. 7a. Considering that GSAM has one more parameter $\alpha$, we plot the accuracy varying with $\alpha$ in Fig. 7b, and show that GSAM consistently outperforms SAM and vanilla training.

## C.2 CONSTANT $\rho$ V.S. DECAYED $\rho_t$ SCHEDULE

Note that Thm. 5.1 assumes $\rho_t$ to decay with $t$ in order to prove the convergence, while SAM uses a constant $\rho$ during training. To eliminate the influence of $\rho_t$ schedule, we conduct ablation study as in Table. 6. The ascent step in GSAM can be applied to both constant $\rho$ or a decayed $\rho_t$ schedule, and improves accuracy for both cases. Without ascent step, constant $\rho$ and decayed $\rho_t$ achieve similar performance. Results in Table. 6 implies that the ascent step in GSAM is the main reason for improvement of generalization performance.

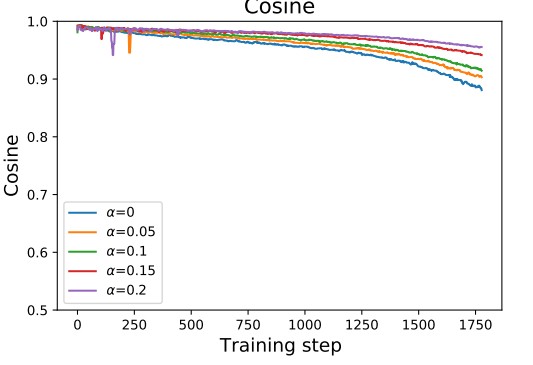

Figure 8: The value of $\cos\theta_t$ varying with training steps, where $\theta_t$ is the angle between $\nabla f(w_t)$ and $\nabla f_p(w_t)$ as in Fig. 2.

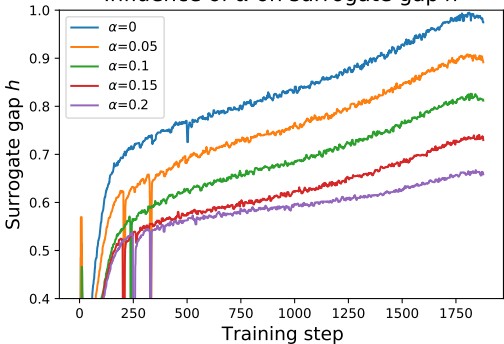

Figure 9: Surrogate gap curve under different $\alpha$ values.

Table 6: Top-1 Accuracy on ViT-B/32 on ImageNet. Ablation studies on constant $\rho$ or a decayed $\rho_t$.

| Vanilla | Constant $\rho$ (SAM) | Constant $\rho$ + ascent | Decayed $\rho_t$ | Decayed $\rho_t$ + ascent |
|---------|----------------------|--------------------------|------------------|---------------------------|
| 72.0 | 75.8 | 76.2 | 75.8 | 76.8 |

### C.3 VISUALIZE THE TRAINING PROCESS

In the proof of Thm. 5.3, our analysis relies on assumption that $\theta_t$ is small. We empirically validated this assumption by plotting $\cos\theta_t$ in Fig. 8, where $\theta_t$ is the angle between $\nabla f(w_t)$ and $\nabla f_p(w_t)$. Note that the cosine value is calculated in the parameter space of dimension $8.8 \times 10^7$, and in high-dimensional space two random vectors are highly likely to be perpendicular. In Fig. 8 the cosine value is always above 0.9, indicating that $\nabla f(w_t)$ and $\nabla f_p(w_t)$ point to very close directions considering the high dimension of parameters. This empirically validates our assumption that $\theta_t$ is small during training.

We also plot the surrogate gap during training in Fig. 9. As $\alpha$ increases, the surrogate gap decreases, validating that the ascent step in GSAM efficiently minimizes the surrogate gap. Furthermore, the surrogate gap increases with training steps for any fixed $\alpha$, indicating that the training process gradually falls into local minimum in order to minimize the training loss.

## D RELATED WORKS

Besides SAM and ASAM, other methods were proposed in the literature to improve generalization: Lin et al. (2020) proposed extrapolation of gradient, Xie et al. (2021) proposed to manipulate the noise in gradient, and Damian et al. (2021) proved label noise improves generalization, Yue et al. (2020) proposed to adjust learning rate according to sharpness, and Zheng et al. (2021) proposed model perturbation with similar idea to SAM. Izmailov et al. (2018) proposed averaging weights to improve generalization, and Heo et al. (2020) restricted the norm of updated weights to improve generalization. Many of aforementioned methods can be combined with GSAM to further improve generalization.

Besides modified training schemes, there are other two types of techniques to improve generalization: data augmentation and model regularization. Data augmentation typically generates new data from training samples; besides standard data augmentation such as flipping or rotation of images, recent data augmentations include label smoothing (Müller et al., 2019) and mixup (Müller et al., 2019) which trains on convex combinations of both inputs and labels, automatically learned augmentation (Cubuk et al., 2018), and cutout (DeVries & Taylor, 2017) which randomly masks out parts of an image. Model regularization typically applies auxiliary losses besides the training loss such as weight decay (Loshchilov & Hutter, 2017), other methods randomly modify the model architecture during training, such as dropout (Srivastava et al., 2014) and shake-shake regularization (Gastaldi, 2017). Note that the data augmentation and model regularization literature mentioned here typically train with the standard back-propagation (Rumelhart et al., 1985) and first-order gradient optimizers, and both techniques can be combined with GSAM.

Besides SGD, Adam and AdaBelief, GSAM can be combined with other first-order gradient optimizers, such as AdaBound (Luo et al., 2019), RAdam (Liu et al., 2019), Yogi (Zaheer et al., 2018), AdaGrad (Duchi et al., 2011), AMSGrad (Reddi et al., 2019) and AdaDelta (Zeiler, 2012).

