# OpenReview forum: "Surrogate Gap Minimization Improves Sharpness-Aware Training"
_ICLR.cc/2022/Conference — ICLR 2022 Poster_

### Official Review · Reviewer_jbS6 · 2021-10-19

**Correctness:** 3
**Technical Novelty And Significance:** 3
**Empirical Novelty And Significance:** 3
**Recommendation:** 6
**Confidence:** 4

**Details Of Ethics Concerns:**

There are no ethics concerns for the paper.

**Main Review:**

The idea of minimizing the surrogate gap while maintaining the perturbed loss with gradient ascent along an orthogonal direction is interesting. The experimental results are comprehensive and seem to be convincing. Here are some issues.

- In the proof of Thm 5.1, the smoothness parameter is $L_p=L+L^2\rho/\epsilon$. Note $\epsilon$ is often very small. Indeed, the default choice of $\epsilon$ is $10^{-12}$. Then the convergence bound would be very crude since $L_p$ is very large, which may not apply well in practice.

- Eq (40) and (41) use inaccurate expressions such as $\approx$. Therefore, these inequalities are not precise. The proof based on these inequalities are therefore not rigorous. Furthermore, in the proof there are some $\beta_{\min}$ and $\beta_{\max}$. These quantities are unknown in practice. They depend on the practical implementations of the algorithm and the authors do not give an upper bound on $\beta_{\max}/\beta_{\min}$. If the term $\beta_{\max}/\beta_{\min}$ is large, the derived bound can be crude. The authors should also be careful about the expectation operators. There are missing expectation operators for Eq (37)-(40), Eq (43)-(54).

- The proof of Thm 5.3 is also not rigorous since the authors use several inaccurate inequalities, e.g. ($\approx$ in (70) and (71))

- In Corollary 5.2.1, the authors tried to show that GSAM can find a smoother minimizer by involving the surrogate gap in the generalization bound. However, this actually does not convey this message. The underlying reason is that both sides of eq (7) involve $C$. Therefore, Corollary 5.2.1 actually gives a bound on the risk in terms of the perturbed training loss if one removes $C$ from both sides: the perturbed training loss is small then the model has a small risk. It does not show the benefit of a small surrogate gap in achieving a better generalization, which is a claim of the paper.

Typos:

$g^{(t)}$ in eq (37) should be $g^{(t)}_\bot$

there are two $++$ in eq (49)

there is a mixing $E_x$ in Thm 5.2

**Summary Of The Paper:**

In this paper,the authors proposed a new method for sharpness aware training. The idea is to note a surrogate gap as a measure of sharpness, which motivates to minimize the perturbed loss and the surrogate gap simultaneously. The authors proposed to use a gradient descent to minimize the perturbed loss and gradient ascent in an orthogonal direction to minimize the surrogate gap. The use of the orthogonal direction is to avoid the influence of the perturbed loss in the ascent step. The paper gives theoretical results to show the connection between the proposed surrogate gap and the sharpness. Convergence rates and generalization bounds are also presented. Experimental results are reported which seem to be convincing.

**Summary Of The Review:**

The idea of minimizing the surrogate gap and the perturbed loss simultaneously is interesting. The experimental results are sufficient. However, the theoretical analysis is not quite convincing.

---

> ### Author Response · Authors · 2021-11-12
> **Thanks for your careful review, we thoroughly address your concerns (Part 2 / 2)**
>
> **Q5. Corollary 5.2.1 does not show the benefit of a small surrogate gap in achieving a better generalization, which is a claim of the paper.**
> Thanks for the very insightful comments. The comment ```The corollary gives a bound on the risk in terms of the perturbed training loss if one removes C from both sides``` is correct. But there is a misunderstanding in the statement ```the perturbed training loss is small then the model has a small risk```: it's only true when $\rho_{train}$ for training equals its real value $\rho_{true}$ determined by the data distribution; in practice, we never know $\rho_{true}$. In the following we show that the minimization of both $h$ and $f_p$ is better than simply minimizing $f_p$ **when $\rho_{true} \neq \rho_{train}$**.
>
> 1. First, we re-write the conclusion of Corollary 5.2.1 as
> $\mathbb{E}_w \mathbb{E}_x f(w,x) \leq f_p + R = C + \widehat{h} + R = C + \rho^2 \sigma / 2  + R + O(\rho^3)$  with probability at least $(1-a)[1-e^{-(\frac{\rho}{\sqrt{2}b}-\sqrt{k})^2}]$,
> where $R$ is the regularization term, $C$ is the training loss, $\sigma$ is the dominant eigenvalue of Hessian. As in lemma 3.3, we perform Taylor-expansion and can ignore the high-order term $O(\rho^3)$. We focus on $f_p = C + \widehat{h}=C+\rho^2 \sigma / 2$ in what follows.
>
> 2. **When $\rho_{true} \neq \rho_{train}$, minimizing $h$ achieves a lower risk than only minimizing $f_p$.**
> (1) Note that after training, **$C$ (training loss) is fixed, but $h$ could vary with $\rho$** (e.g. when training on dataset A and testing on an unrelated dataset B, the training loss remains unchanged, but the risk would be huge and a large $\rho$ is required for a valid bound).
> .
> (2) With an example, we show a low $f_p$ is insufficient for generalization, and a low $\sigma$ is necessary.
>
> >* Suppose we use $\rho_{train}$ for training, and consider two solutions with $C_1, \sigma_1$ (SAM) and $C_2, \sigma_2$ (GSAM). Suppose they have the same $f_p$ during training for some $\rho_{train}$, so
> $f_{p1}=C_1 + \sigma_1 / 2 \times \rho_{train}^2 = C_2 + \sigma_2 / 2 \times \rho_{train}^2=f_{p2}$
> Suppose $C_1 < C_2$ so $\sigma_1 > \sigma_2$.
> >* When $\rho_{true} > \rho_{train}$, we have
> $\texttt{risk bound 1}= C_1 + \sigma_1 /2 \times \rho_{true}^2 + R > \texttt{risk bound 2} = C_2 + \sigma_2 / 2 \times \rho_{true}^2 + R$
> This implies that a **small $\sigma$ helps generalization**, but only a low $f_{p1}$ (caused by a low $C_1$ and high $\sigma_1$) is insufficient for a good generalization.
> >* Note that $\rho_{train}$ is fixed during training, so minimizing $h_{train}$ during training is equivalently minimizing $\sigma$ by Lemma 3.3
>
> 3. **Why we are often unlucky to have $\rho_{true}>\rho_{train}$**
> (1) First, the test sets are almost surely **outside** the convex hull of the training set because ```interpolation almost surely never occurs in high-dimensional (>100) cases``` ( [1] by Balestriero and Yann LeCun et al.). As a result, the variability of (train + test) sets is almost surely larger than the variability of (train) set. Since $\rho$ increases with data variability (see point 4 below), we have $\rho_{true} > \rho_{train\\\_set}$ almost surely.
> .
> (2) Second, we don’t know the value of $\rho_{true}$ and can only guess it. In practice, we often guess a small value because training often diverges with large $\rho$ (as observed in [3,4]).
>
> 4. **Why $\rho$ increases with data variability.**
> In Corollary 5.2.1, we assume weight perturbation $\delta \sim \mathcal{N}(0, b^2 I^k)$. The meaning of $b$ is the following. If we can randomly sample a fixed number of samples from the underlying distribution, then training the model from scratch (with a fixed seed for random initialization) gives rise to a set of weights. Repeating this process, we get many sets of weights, and their standard deviation is $b$. Since the number of training samples is limited and fixed, the more variability in data, the more variability in weights, and the larger $b$.
> Note that Corollary stated that the bound holds with probability proportional to $[1-e^{-(\frac{\rho}{\sqrt{2}b}-\sqrt{k})^2}]$. In order for the result to hold with a fixed probability, $\rho$ must stay proportional to $b$, hence $\rho$ also increases with the variability of data.
>
> Finally, we request a re-evaluation of the paper as a whole since we felt it was disproportionately penalized for the theoretical part. Reviewer ianU commented “this is a strong paper even without the theoretical analysis”. Our responses above address all concerns about the theoretical part. Please let us know any further questions.
>
> [1] Balestriero et al. Learning in High Dimension Always Amounts to Extrapolation
> [2] Anonymous. UNDERSTANDING SHARPNESS-AWARE MINIMIZATION
> [3] Foret, et al. Sharpness-aware minimization for efficiently improving generalization
> [4] Chen et al. When vision transformers outperform resnets without pretraining or strong data augmentations

---

> > ### Comment · Reviewer_jbS6 · 2021-11-27
> > **thank you for your response**
> >
> > Thank you for your detailed response. I am satisfied with the revision and the response. I will increase my score.

---

> > > ### Author Response · Authors · 2021-11-30
> > > **Thanks for your review**
> > >
> > > Thanks for your review, we are glad to resolve your concerns and appreciate your advice on the improvement of our paper.

---

> ### Author Response · Authors · 2021-11-12
> **Thanks for your careful review, we thoroughly address your concerns (Part 1 / 2)**
>
> Thanks for your review, we carefully address your concerns below.
>
> **Q1. the convergence bound would be very crude since $L_p$ is very large, which may not apply well in practice.**
> 1. Thanks for pointing this out. It is due to the proof techniques and can be fixed. We have added appendix A.7, a proof for the convergence w.r.t. $f_p$ without relying on the L-smoothness of $f_p$, and the convergence rate is still $O(\log T / \sqrt{T})$. We also write out the constant terms in Eq.96 and 97, and all constants can be well-bounded.
> 2. In the proof of Thm 5.1, only step 1 requires the $L_p$ term (and steps 2 and 3 are not affected). Therefore, combined with (1), our result in Thm 5.1 still holds.
>
> **Q2. $\approx$ in proof**
> Thanks for pointing this out. We have fixed this in the updated version. We used $\approx$ when we ignore the higher-order terms in Taylor expansion.
>
> 1. For example, Eq.40 and Eq.71 rely on the following Taylor-expansion
>   $\operatorname{tan} x = x + O(x^2), \operatorname{sin} x = x + O(x^2), x \to 0$
> which are standard results in calculus (https://socratic.org/questions/what-is-the-limit-as-x-approaches-0-of-tanx-x)
>
> 2. Eq.31 and Eq.70 rely on Taylor expansion on the gradient:
> $\nabla f(w+\delta) = \nabla f(w) + H(w) \delta + O(\vert \vert \delta \vert \vert^2)$
>
> 3. In the new version, we add higher-order terms and use $=$. Please note that this does not affect the rigorousness of proof, as the low-order terms have non-zero coefficients and are the dominant terms.
>
> 4. If we want to be extremely strict, we can multiply the coefficients of low-order terms by a constant slightly larger than 1 (e.g. 1.1). For example, $O(x^2) < 0.1 |x|, x \to 0$, then we can safely remove the high-order term $O(x^2)$. This would only affect the constants rather than the order in the risk bound, and the final bound would be slightly loose but a strictly rigorous upper bound.
>
> **Q3. The bound on $\beta_{max} / \beta_{min}$**
> 1. A geometric interpretation is that $\beta = \operatorname{cos} \theta$ where $\theta$ is the angle between $\nabla f(w)$ and $\nabla f_p(w)$. In theory, $\theta_t$ is upper bounded by $\rho_t L / G$, so $\beta_{min} = \operatorname{cos} \theta_t^{max}$ is lower bounded by $\sqrt{ 1 - \operatorname{sin}^2(\frac{\rho_t L}{G}) } = \sqrt{ 1 - (\frac{\rho_t L}{G})^2 + O(\rho_t)^2 }$, which is very close to 1 by controlling $\rho_t$. $\operatorname{cos}$ is naturally upper-bounded by 1. Hence, the term $\beta_{max}/\beta_{min}$ is upper bounded by $1/\sqrt{ 1 - (\frac{\rho_t L}{G})^2}$ and lower bounded by 1 --- this range can be very tight by controlling $\rho_t$.
>
> 2. We plot its value in Fig.8 in appendix C. Even the smallest value (for SAM) is larger than 0.85. For GSAM, the minimum is above 0.9, and $\beta_{max} \leq 1$, so the term $\beta_{max} / \beta_{min}$ is well-bounded (between 1 and 1/0.85) in the empirical study.
>
> **Q4. Missing expectation symbols**
> Thanks for pointing it out. We have added symbols to $\mathbb{E} g_t$ and $\mathbb{E} \vert \vert d_t \vert \vert^2$. Note that we are performing expectations conditioned on observations up to step $t$, so only $g_t, d_t, w_{t+1}$ are random variables and need an expectation symbol. This follows a standard trick in the convergence proof.
>
> [1] Randall Balestriero, Jerome Pesenti, and Yann LeCun. "Learning in High Dimension Always Amounts to Extrapolation." arXiv preprint arXiv:2110.09485 (2021).
> [2] Understanding Sharpness-Aware Minimization
> [3] Foret, Pierre, et al. "Sharpness-aware minimization for efficiently improving generalization." arXiv preprint arXiv:2010.01412 (2020).
> [4] Xiangning Chen et al. When vision transformers outperform resnets without pretraining or strong data augmentations

---

### Official Review · Reviewer_6UYP · 2021-10-29

**Correctness:** 3
**Technical Novelty And Significance:** 3
**Empirical Novelty And Significance:** 3
**Recommendation:** 8
**Confidence:** 4

**Main Review:**

Strengths

+ I really like the idea of this paper that notices that both $f_p$ and $h$ are crucial in boosting's SAM's success.

+ The empirical findings are nice to see.

+ The theoretical findings are also nice to have, but perhaps not as convincing as the empirical findings.

Weaknesses

- I perused some of the proofs of the theoretical results. For the most part, they look fine. However, the authors can be rather sloppy at times. One cannot use $\approx$ in a proof without quantifying what it means. For example, in the transition from (39) to (40), the authors justify the change of $\sin\theta_t$ to $\tan\theta_t$ by mentioning that $\theta_t$ is small. I believe that the authors would do better if they quantified the remainder terms carefully.

- One other deficiency is that the performance plots/tables do not include error bars.

**Summary Of The Paper:**

This paper builds on the main idea of SAM. Namely, the authors noticed that minimizing the perturbed loss $f_p$ is insufficient for guaranteeing flat minima. The authors then defined the notion of surrogate gap $h$, which they propose to minimize together with the perturbed loss. By using GSAM which consists of two gradient descent/ascent steps that include the minimizing $f_p$ and $h$, the authors show theoretically and experimentally that this results in a model with better generalization than SAM.

**Summary Of The Review:**

The idea is nice. The empirical performances look promising and the theoretically analysis generally sound (though sloppy). I recommend a weak accept.

---

> ### Author Response · Authors · 2021-11-12
> **Thanks for your review, we address your concerns below**
>
> Thanks for your review, we carefully address your concerns below.
>
> **Q1. Remainder terms and $\approx$ in proof**
>
> Thanks for pointing this out. We have fixed this in the updated version. We use $\approx$ when we ignore the higher-order terms in Taylor expansion.
>
> For example, Eq.40 and Eq.71 rely on the following Taylor-expansion
>   $\operatorname{tan} x = x + O(x^2), \operatorname{sin} x = x + O(x^2), \operatorname{sin} x = \operatorname{tan} x +O(x^2), x \to 0$
> which are standard results in calculus (https://socratic.org/questions/what-is-the-limit-as-x-approaches-0-of-tanx-x)
>
> Eq.31 and Eq.70 rely on Taylor expansion on the gradient:
> $\nabla f(w+\delta) = \nabla f(w) + H(w) \delta + O(\vert \vert \delta \vert \vert^2)$
>
> In the new version, we added higher-order terms and used $=$. Please note that this does not affect the rigorousness of proof, as the low-order terms have non-zero coefficients and are the dominant terms.
>
> **Q2 Error bars of results**
> Thanks for your comments. As stated in Sec 6.1 paragraph 2, we have tested the standard deviation in table 1 and found them smaller than 0.1% across 3 independent runs, which is negligible compared to the improvements.

---

> > ### Comment · Reviewer_6UYP · 2021-11-23
> > **Thanks for the response**
> >
> > Thanks for the response. This is a good paper. I have upped my score to 8.

---

> > > ### Author Response · Authors · 2021-11-24
> > > **Thanks for your review!**
> > >
> > > Thanks for your review. We appreciate your suggestions to improve our paper!

---

### Official Review · Reviewer_ianU · 2021-11-02

**Correctness:** 3
**Technical Novelty And Significance:** 3
**Empirical Novelty And Significance:** 3
**Recommendation:** 6
**Confidence:** 4

**Main Review:**

### Strengths
* The paper is well written, and I like the clear motivations and illustrations shown in Figure 1-3.
* The proposed method is simple, elegant and well-motivated yet appears to work extremely well across a wide range of experiments.

### Weaknesses
* The main optimization objective presented at the beginning of Section 4.1 is not entirely clear to me and seems not mathematically rigorous. The quantity (f_p(w), h(w)) is a vector, so is this to be understood as a multi-objective optimization task?  What is the definition of a solution of this problem?
* Later it is mentioned that GSAM solves min(f_p, h), but it is not clear whether the convergence theorem actually shows that GSAM finds a "solution" of this objective.  Instead of the decomposition of the gradient into two components, one could for example consider an algorithm which directly computes gradients of the min(f_p, h)-function.
* I am a bit unsure whether "surrogate gap" is a good name for h(w). Isn't h(w) simply the sharpness as considered eg. in Foret et al?
* An important aspect of SAM is what is called "m-sharpness" in Foret et al, i.e., computing the perturbed loss on mini batches.   I could not find any mentioning of the "m"-parameter in the paper, and since the performance of SAM greatly depends on it, it would be great to confirm that a reasonable value is used in this work.
* Overall, I am not sure whether the paper really needs to include the theoretical results presented in Section 5.2.  They appear very hard to digest for the reader and I am unsure whether anything can be concluded from it. I believe this is a strong paper even without the theoretical analysis.

**Summary Of The Paper:**

The paper presents a new method for improving generalization in deep learning called GSAM which builds upon and extends recent works on adversarial weight perturbation and sharpness-aware minimization (SAM). The paper argues that minimizing the loss function used in SAM (which replaces the loss by its maximum in a neighbourhood) does not always lead to flat minima. Based on this insight, the paper puts forward the following contributions:
1. A new loss function which does not suffer from these problems, along with a new algorithm derived from the loss function
2. Theoretical results (generalization bounds and convergence theorems) supporting the algorithm and new loss function
3. An extensive numerical evaluation, showing that the proposed GSAM method leads to significant gains over SAM


**Summary Of The Review:**

The paper has some smaller weaknesses (see main review) but if these are addressed in the rebuttal I am inclined to increase my score and recommend acceptance of the paper.  Overall, the work proposes a simple and elegant method, which is well motivated and gives state-of-the-art performance in an extensive numerical evaluation.  Therefore, I believe it will be of great interest to the ICLR community.

---

> ### Author Response · Authors · 2021-11-12
> **Thanks for your review, we have carefully addressed your comments**
>
> Thanks for your review, we have carefully addressed your comments below.
>
> **Q1. so is this to be understood as a multi-objective optimization task? What is the definition of a solution of this problem?**
> 1. Yes —we will revise the paper and explicitly call out the name of multi-objective optimization earlier. The strict definition of the solution to multi-objective optimization is called “Pareto optimality” as defined in [1]. The exact solution is complicated and out of the scope of our paper, but intuitively, a solution is “Pareto optimal” when no other solution can beat it in all tasks. However, a “Pareto optimal” solution does not necessarily be the best for all individual tasks.
>
> 2. Just like it is hard to get the global minimum for single-objective optimization in the stochastic non-convex case, it is even harder to get the Pareto optimal for multiple-objective stochastic non-convex optimization. Most theoretical analysis only shows the algorithm converges to a stationary point in the single-objective case, and our results show GSAM finds a solution that is stationary for every task in the multi-objective case. (see Q2 below for details)
>
> **Q2. it is not clear whether the convergence theorem actually shows that GSAM finds a "solution" of this objective.**
> 1. Even for single-objective optimization in the non-convex stochastic case, most proofs only guarantee that the algorithm converges to a “stationary point” [2][3], meaning that $\mathbb{E} \vert \vert \nabla f(w_t) \vert \vert^2 \to 0$ as $t \to \infty$. There is no guarantee that the final solution is the global minimum with the lowest training loss.
>
> 2. Similarly, for multi-objective optimization, our proof guarantees GSAM finds a solution that is a stationary point for each objective, i.e. $\mathbb{E} \vert \vert \nabla f(w_t) \vert \vert^2 \to 0$ as $t \to \infty$ and $\mathbb{E} \vert \vert \nabla f_p(w_t) \vert \vert^2 \to 0$ as $t \to \infty$ and $\mathbb{E} \vert \vert \nabla h(w_t) \vert \vert^2 \to 0$ as $t \to \infty$. Hence, from the typical convergence criteria for stochastic non-convex optimization, we have found a solution.
>
> **Q3. one could for example consider an algorithm which directly computes gradients of the min(f_p, h)-function.**
> The gradients of the two objectives $f_p$ and $h$ often compete, making the problem inherently hard as stated in [1]. [1] proposed a method that requires internal iterations within each update step, which is very time-consuming and impractical for large models. We are not aware of existing methods that can easily solve this problem.
>
> **Q4. Isn't h(w) simply the sharpness as considered eg. in Foret et al?**
> Mathematically, our “surrogate gap” is the “sharpness” in the SAM paper. However, we believe the word “sharpness” should refer to some inherent attributes of the loss landscape of a trained model, and should be directly comparable. The value of $h$ depends on $\rho$, and $h_1 < h_2$ does not imply model 1 has a flatter surface than model 2 if $\rho_1$ and $\rho_2$ are different. Therefore, we stick to using “sharpness” for inherent metrics such as the dominant eigenvalue of Hessian, and use “surrogate gap” for $h$.
>
> **Q5. the "m"-parameter in the paper**
> We follow the settings in [4] with a batchsize of 4096 and 128 TPU cores for training. Each core receives m=32 images. The SAM paper reported a small m typically improves performance, such as m=1,4,16. Our performance could be further improved with a smaller m. To make the paper self-contained, we will include these details in the revised version so that readers do not need to check [4] for them.
>
> **Q6. the theoretical results presented in Section 5.2**
> We feel this section (about theoretical results) complements the empirical results by revealing some properties of our algorithm (e.g., generalization). Nonetheless, we agree with the reviewer that the other sections are more important especially for the practical use of GSAM, so we made this section concise. We briefly summarize the take-home message for Sec 5.2 below.
>
> 1. Thm5.2 and Corollary 5.2.1 together state that, the loss on the test set can be upper-bounded by the sum of training loss, surrogate gap (proportional to the dominant eigenvalue of Hessian), and a regularization term.
>
> 2. Thm 5.3 states that the surrogate gap by GSAM is smaller than SAM by at least a constant, so the loss surface by GSAM is guaranteed to be flatter and the generalization is better.
>
>
> [1] Sener, Ozan, and Vladlen Koltun. "Multi-task learning as multi-objective optimization." arXiv preprint arXiv:1810.04650 (2018).
> [2] Reddi, Sashank J., Satyen Kale, and Sanjiv Kumar. "On the convergence of adam and beyond." arXiv preprint arXiv:1904.09237 (2019).
> [3] Jain, Prateek, and Purushottam Kar. "Non-convex optimization for machine learning." arXiv preprint arXiv:1712.07897 (2017).
> [4] Xiangning Chen et al. When Vision Transformers Outperform ResNets without Pre-training or Strong Data Augmentations

---

### Official Review · Reviewer_jYWK · 2021-11-09

**Correctness:** 3
**Technical Novelty And Significance:** 3
**Empirical Novelty And Significance:** 3
**Recommendation:** 6
**Confidence:** 3

**Main Review:**

Strengths
1. This submission provides a more reasonable way to find the flatten minima rather than the heuristic way utilized in SAM.
2. With negligible additional computation, this method improves the generalization performance.

Weaknesses
1. It is well known that SAM-based methods may lose their superior on some more complex DNN architecture, such as EffeicientNet for CNN-based model, Swin for vision transformer. This submission still performs the experiments on the vanilla architectures as the SAM did. We may wish the more exciting results on the advanced DNNs.
2. Eq.5 seems to do not fit the updating rule provided in this submission.  Although the authors alert the potential caveat for optimizing $f_p + \lambda h$, the proposed GSAM still aims at minimizing this objective. A minor wise modification is to exclude the component in the gradient of the second term, which may degenerate the values of the first term. Hence, the convergence guarantee may require more efforts to deal with the proposed modification; however, no more discussions are given.
3. The convergence guarantee is a classical conclusion for non-convex stochastic optimization. If the authors cannot provide the details that show how to treat $f_p$ technically, there is no need to provide the guarantee in the main part, which cannot offer any superiority of GSAM for readers. So does the generalization guarantee, which is a direct application of general PAC-Bayesian bound. Thm. 5.3  is more interesting than the previous ones.


**Summary Of The Paper:**

It is a good modification to the SAM method, which may further improve the performance.

**Summary Of The Review:**

It is a good submission that provides an interesting way to improve the updating rule of SAM. However, some modifications for the technical part of this submission are still needed.

---

> ### Public Comment · ~Jean_Kaddour1 · 2021-11-10
> **Ask for reference showing that "SAM-based methods may lose their superiority"**
>
> ___It is well known that SAM-based methods may lose their superior on some more complex DNN architecture, such as EffeicientNet for CNN-based model, Swin for vision transformer.___
> This is very interesting. So far, I only read that SAM works well for vision transformers [1]. Can you please share a reference on this?
> [1] Chen, Xiangning, Cho-Jui Hsieh, and Boqing Gong. "When Vision Transformers Outperform ResNets without Pretraining or Strong Data Augmentations." arXiv preprint arXiv:2106.01548 (2021).

---

> ### Public Comment · ~Juntang_Zhuang1 · 2022-01-31
> **Thanks for your review, we carefully address your comments**
>
> Thanks for your comments, we carefully address your concerns below.
>
> **Q1. SAM-based methods may lose their superior on some more complex DNN architecture**
> 1. Thanks for bringing this up. Could you provide any references on this? To the best of our knowledge, we are not aware of the degradation of SAM's performance on EfficientNet or Swin Transformer. In fact, we found the original paper of SAM [1] reported improved results with SAM on EfficientNet.
>
> 2. We notice that the training for EfficientNet [2] and Swin Transformer [3] uses much heavier data augmentations than usual, hence the optimal $\rho$ should be different from reported in [1] under light augmentation, which could be the reason for performance degradation (if it exists). The same phenomenon is also observed in [4] and in our experiments in Sec 6.4, where we empirically validate that GSAM still achieves improvements under light to strong augmentations.
>
> **Q2. Eq.5 seems to do not fit the updating rule provided in this submission.**
> Sorry but we believe this is a misunderstanding of our paper, our theoretical analysis strictly follows Algo.1, which is a solution to Eq.5. We will update the draft to make this point more clear. Please see details below:
>
> 1. GSAM avoids the caveat because it does **not** aim to solve $f_p+\lambda h$; instead, it optimizes Eq.5, which is a *multi-objective* optimization problem. Please note that the caveat is **not the goal** to minimize the two objectives of $f_p$ and $h$ in Eq.5, but **how** to minimize them.
> In the paper, we discussed two ways to minimize $f_p$ and $h$: **(a)** directly combining their gradients, $\nabla f_p + \lambda \nabla h$, and descending in that direction, **(b)** GSAM in Algo.1. Figure 2 and Table 2 validate that (b) gives rise to better solutions to Eq.5 than (a) does. Moreover, we discuss the difference between (a) and (b) below.
>
> * Directly taking gradient descent in the direction of $\nabla f_p + \lambda \nabla h$ is invalid as stated in the abstract of [5] and Section 4 in our paper because the two gradients directions, $\nabla f_p$ and $\lambda \nabla h$, are very likely to have negative inner-product. On the contrary, GSAM avoids these conflicts by updating in the orthogonal direction, with the final update direction of $\nabla f_p + \alpha \nabla_\perp f$.
>
> * $\nabla f_p + \lambda \nabla h \neq \nabla f_p + \alpha \nabla_\perp f$ even if we tune the scalars $\lambda$ or $\alpha$ (unless set $\lambda=\alpha=0$). The two methods are hence different by nature because $\nabla h$ and $\nabla_\perp f$ point to different directions.
>
> 2. Excluding $\lambda \nabla h$ is equivalent to $\operatorname{min} f_p$, which reduces to SAM --- Please correct us if we misunderstood "A minor wise modification is to exclude the component in the gradient of the second term" in the review.
>
> **Q3. The convergence guarantee is a classical conclusion for non-convex stochastic optimization.**
> Sorry but this comment might be due to a misunderstanding. We have to provide new proofs because the classical regime does not directly hold for our work for the following reasons.
>
> 1. The classical optimization with first-order gradient information has an important condition: the update follows the gradient of the loss function (or dimension-wise rescaled gradient for adaptive optimizers). This implies that, when the stepsize is infinitely small and the algorithm runs for infinitely long, the trajectory is the gradient flow.
>
> 2. However this condition is **violated** by GSAM, because: **(1)** the update direction has a perpendicular direction component, so the trajectory is guaranteed to deviate from the gradient flow of the loss function, as shown in Fig.3. **(2)** there is no way to write the update of GSAM as following the gradient of $f, f_p$ or the exact gradient of any function of $f$ and $f_p$. Therefore, GSAM does not fall into the conventional regime, and classical results do not directly apply to GSAM.
>
> 3. For the techniques of the proof, GSAM has a term of $\langle \nabla f(w_t), \nabla_\perp f(w_t) \rangle$ in Eq.38 caused by orthogonal update ($\nabla_\perp f(w_t)$ vertical to $\nabla f_p(w_t)$ not $\nabla f(w_t)$), which does not exist in the classical optimization, and the biggest efforts in our convergence proof is to bound the influence of this term.
>
>
> [1] Pierre Foret et al. Sharpness-aware Minimization for Efficiently Improving Generalization
> [2] Tan, Mingxing, and Quoc Le. "Efficientnet: Rethinking model scaling for convolutional neural networks." International Conference on Machine Learning. PMLR, 2019.
> [3]  Liu, Ze, et al. "Swin transformer: Hierarchical vision transformer using shifted windows." arXiv preprint arXiv:2103.14030 (2021).
> [4] Xiangning Chen et al. When vision transformers outperform resnets without pre-training or strong data augmentation
> [5] Sener, Ozan, and Vladlen Koltun. "Multi-task learning as multi-objective optimization." arXiv preprint arXiv:1810.04650 (2018).

---

### Decision · Program_Chairs · 2022-01-20

**Decision:**

Accept (Poster)

**Comment:**

The paper proposes an interesting and well-motivated improvement of Sharpness Aware Minimization.  Overall the AC and reviewers are satisfied by the author feedback in improving the solidity and rigor of the theoretical results.

The points made by the authors in response to the reviewers initial concerns are essential, especially those regarding interpretation of Corollary 5.2.1, making the proofs rigorous, and fixing the potential for crude convergence bounds. It is therefore critical that the authors incorporate them into their manuscript.